# Multi-domain O-GlcNAcase structures reveal allosteric regulatory mechanisms

Sara Basse Hansen [1,6], Sergio G. Bartual[1,6], Huijie Yuan[1,2], Olawale G. Raimi[2], Andrii Gorelik [2,5], Andrew T. Ferenbach[1], Kristian Lytje[3,4], Jan Skov Pedersen [3,4], Taner Drace[1,3], Thomas Boesen [1,3] & Daan M. F. van Aalten [1,2] ✉

Nucleocytoplasmic protein O-GlcNAcylation is a dynamic modification catalysed by O-GlcNAc transferase (OGT) and reversed by O-GlcNAc hydrolase (OGA), whose activities are regulated through largely unknown O-GlcNAc-dependent feedback mechanisms. OGA is a homodimeric, multi-domain enzyme containing a catalytic core and a pseudo-histone acetyltransferase (pHAT) domain. While a catalytic structure has been reported, the structure and function of the pHAT domain remain elusive. Here, we report a crystal structure of the *Trichoplax adhaerens* pHAT domain and cryo-EM data of the multi-domain *T. adhaerens* and human OGAs, complemented by biophysical analyses. Here, we show that the eukaryotic OGA pHAT domain forms catalytically incompetent, symmetric homodimers, projecting a partially conserved putative peptide-binding site. In solution, OGA exist as flexible multi-domain dimers, but catalytic core-pHAT linker interactions restrict pHAT positional range. In human OGA, pHAT movements remodel the active site environment through conformational changes in a flexible arm region. These findings reveal allosteric mechanisms through which the pHAT domain contributes to O-GlcNAc homeostasis.

Nucleocytoplasmic modification of 1000s of proteins with O-linked *N*-acetylglucosamine (O-GlcNAc) is a dynamic process[1–3] and modulates a broad range of critical cellular processes such as metabolism, translation, stress response, transcription and protein homoeostasis[3,4]. O-GlcNAcylation is mediated by a single pair of enzymes: the O-GlcNAc transferase (OGT) that adds O-GlcNAc to serine and threonine residues, and the O-GlcNAcase (OGA), which removes O-GlcNAc from the modified proteins[2,4,5]. This modification is particularly critical for the development of the central nervous system in vertebrates, and de novo missense variants in *OGT* and *OGA* have been associated with neurodevelopmental disorders[6–8]. In metazoans, the O-GlcNAc system is conserved across all taxa[9], including the simplest multicellular animal known to possess a functional O-GlcNAc system, the metazoan *Trichoplax adhaerens*[10]. *T. adhaerens* possesses two *OGA* genes producing two functional proteins, a shorter *Ta*OGA53, lacking the C-terminal domain, and a longer *Ta*OGA54 (the latter referred to as *Ta*OGA throughout this manuscript)[10], whereas vertebrates possess several OGA isoforms produced by alternative splicing of a single *OGA* gene (also known as "meningioma expressed antigen 5" (*mgea5*)[11].

The human full-length OGA (hOGA), considered to be the primary isoform, is found in the nucleocytoplasmic space as an obligate dimer and is composed of three domains (Fig. 1a). The N-terminal catalytic domain belongs to the CAZy GH84 glycosyl hydrolase family and

[1]Section for Neurobiology and DANDRITE, Department of Molecular Biology and Genetics, Aarhus University, Aarhus, Denmark. [2]School of Life Sciences, University of Dundee, Dundee, UK. [3]The Interdisciplinary Nanoscience Center (iNANO), Aarhus University, Aarhus, Denmark. [4]Department of Chemistry, Aarhus University, Aarhus, Denmark. [5]Present address: Sir William Dunn School of Pathology, University of Oxford, Oxford, UK. [6]These authors contributed equally: Sara Basse Hansen, Sergio G. Bartual. ✉e-mail: daan@mbg.au.dk

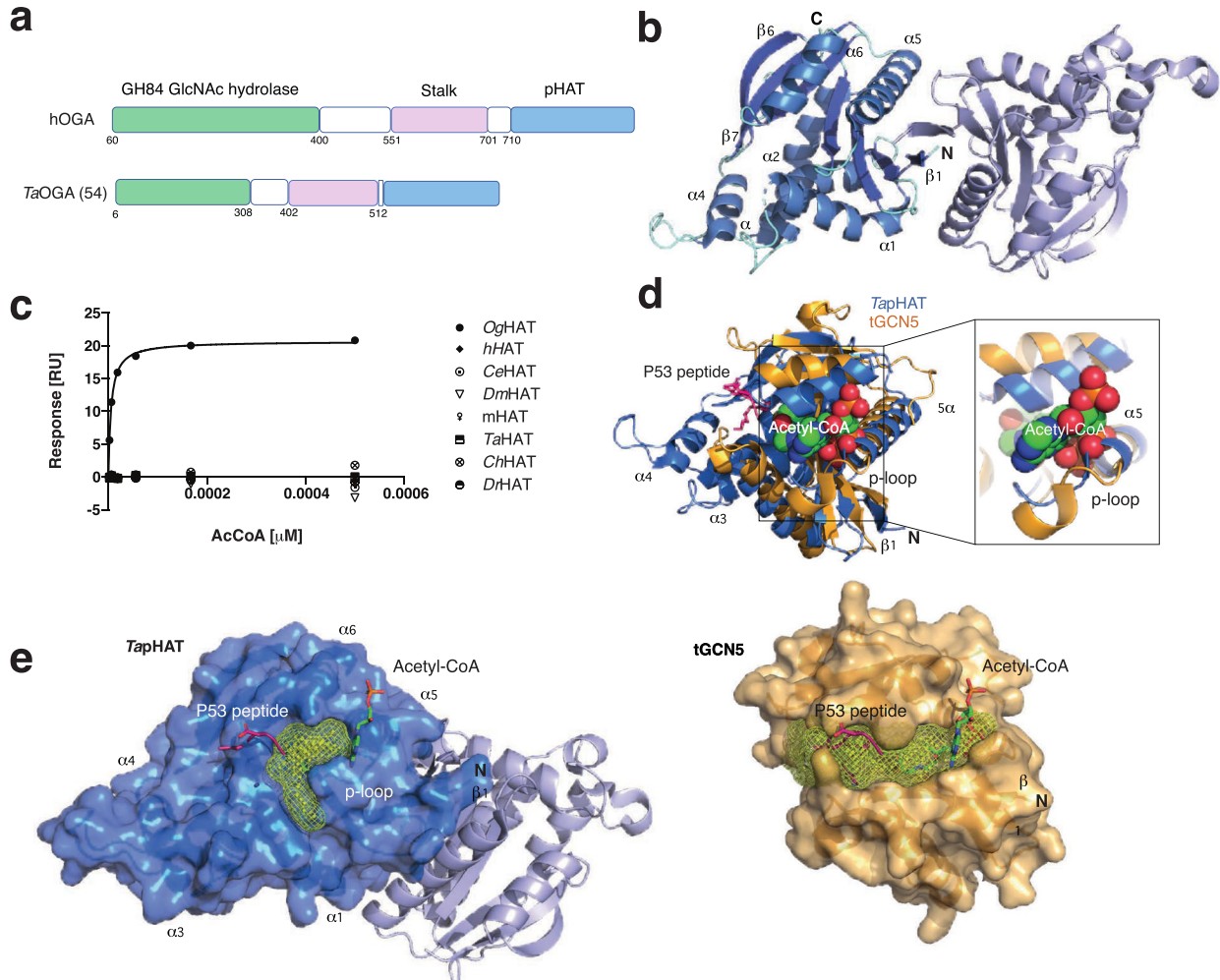

**Fig. 1 | Structure and conservation of the *Ta*pHAT domain dimer. a** Schematic representation of hOGA and *Ta*OGA domain organisation. The white sections represent regions of predicted disorder, the GH84 hydrolase domain is represented in green, while the pHAT domain is coloured in light blue. **b** Cartoon representation of *Ta*pHAT domain structure. Monomer A α-helices are coloured in marine blue, the β-sheets in blue, while the unstructured regions are coloured aquamarine. Monomer B is coloured in slate. **c**) SPR experiment showing acetyl-CoA (AcCoA) binding by *Og*AT (positive control) but not by metazoan proteins (*h*=human, *Ce*=*C. elegans*, *Dm*=*D. melanogaster*, *m*=mouse, Ta *T. adhaerens*, *Ch*=chicken, *Dr*=*D. rerio*). **d** Superposition of the *Ta*pHAT domain (marine) with the tGCN5 structure (PDB 1Q2D[38], bright orange) co-crystallised with acetyl-CoA (spheres) and a P53 peptide (hot pink). A detailed view of the acetyl-CoA putative binding is depicted on the right. *Ta*pHAT domain structural elements are annotated for easy reference. **e** *Ta*pHAT domain dimer overlayed with the ligands extracted from the tGCN5 structure (left). Only monomer A surface is shown (marine). The surface representation of the tGCN5 structure (bright orange) is represented on the right for comparison. Cavities were detected with the PyMOL plugin CavitOmix[83] and represented as yellow meshes in both structures.

employs a substrate-assisted catalytic mechanism to hydrolyse O-GlcNAc from target proteins[12–14]. This active site is a target for competitive inhibitors designed to increase O-GlcNAc levels, offering possible therapeutic potential for neurodevelopmental diseases such as Alzheimer's and Parkinson's disease[15,16]. Crystal structures of a truncated hOGA lacking the C-terminal domain have revealed a homodimer, in which the helical stalk domain contributes to the dimer interface[17–19]. More recently, a cryo-EM structure of the hOGA catalytic core in complex with human OGT has been reported[20]. Alongside earlier studies of substrate complexes with a bacterial OGA orthologue[21], these hOGA structures reveal that substrate recognition is shaped by regions beyond the active site. For instance, the stalk domain, which consists of a helical bundle and an 'arm region' (residues 665–685) close to the active site, plays a crucial role in shaping substrate accessibility by contributing to the formation of the peptide-binding cleft that is capable of binding glycopeptides in a bidirectional mode[19]. Mutations in the stalk domain, such as S652F, can alter hOGA activity and have been implicated in cancer progression by enhancing

O-GlcNAc removal from PDLIM7, ultimately suppressing p53 gene expression and promoting malignancy[22].

While the OGA C-terminal domain (Fig. 1a) is predicted to adopt a fold similar to the GCN5 histone *N*-acetyltransferase family and was initially reported to possess histone acetyltransferase activity in vitro[23], subsequent studies failed to support this[24,25]. The domain is now regarded as a non-catalytic pseudo-histone acetyltransferase (pHAT) domain of unknown function. Nevertheless, mutational evidence implicates that the pHAT domain is involved in the regulation of acetylation processes. For instance, the Y891F mutation in the pHAT domain has been reported to impair the acetylation of PKM2 in cells, yet purified OGA does not directly acetylate PKM2. This has led to the hypothesis that the pHAT domain acts as a scaffold, recruiting an unknown acetyltransferase[26]. These findings suggest that, rather than retaining catalytic function, the pHAT domain may have evolved to mediate specific protein-protein interactions and/or contribute to the allosteric regulation of OGA. Interestingly, a missense *OGA* variant in the pHAT domain was recently associated with neurodevelopmental

delay[8], further highlighting the importance of this domain. *OGT* missense variants leading to intellectual disability are associated with a compensatory loss of OGA mRNA and protein, raising the intriguing possibility that the resulting loss of the pHAT-associated functions may also contribute to the mechanisms underpinning this disease.

The pHAT domain has been reported to be involved in the translocation of hOGA to the nucleus in response to DNA damage[27], where it modulates transcription by altering the O-GlcNAcylation patterns on transcription factors, RNA polymerase II and nucleosomes[28–30]. However, it is not known how the pHAT domains orchestrate this response, or whether it is required for interactions with chromatin-associated proteins. To maintain O-GlcNAc homeostasis, OGA is known to be subjected to multiple layers of regulation. Transcriptional and translational feedback mechanisms adjust *OGA* expression in minutes to hours in response to cellular needs[31]. However, more immediate regulation on shorter time scales could also be achieved through post-translational modifications such as Ser405 O-GlcNAcylation[32] and/or allosteric effects that could involve the non-catalytic domains of OGA, such as the pHAT domains[33]. In summary, the function and structural position of the pHAT domains relative to the OGA homodimeric catalytic core, and their exact contribution to OGA regulation remain unclear.

Here, we report the crystal structure of the isolated pHAT domain homodimer and the cryo-EM full-length homodimeric structure of *Ta*OGA. SPR experiments demonstrate that while the pHAT domains in eukaryotic OGAs retain a putative acceptor binding site, they cannot bind acetyl-CoA, classifying them as pseudo-histone acetyl transferases. In *Ta*OGA, negatively charged linkers appear to decouple the pHAT domains from the symmetric homodimeric catalytic core, allowing them to adopt flexible and undefined positions. In contrast, cryo-EM structures of the multi-domain hOGA homodimer reveal that its pHAT domains are flexibly tethered via proline-rich linkers that disrupt the two-fold symmetry of the catalytic core. Although these domains exhibit conformational variability, they remain partially associated with the core and are sufficiently ordered to be defined, but not fully resolved, in the cryo-EM maps. This asymmetry restricts hOGA to a range of conformations in which the putative peptide binding sites on the pHAT domain become exposed. These observations were further supported by small-angle X-ray scattering (SAXS) experiments. Notably, the positioning of the pHAT domains influences the wider active site environment via key conformational changes in the flexible arm region. Together, the catalytic core-pHAT linker and presence of the pHAT domain modulate O-GlcNAc homeostasis. Collectively, these multi-domain OGA structures reveal previously unknown molecular mechanisms of allosteric regulation.

## Results

### The isolated pHAT domains of TaOGA and hOGA form homodimers

Although hOGA is a multi-domain protein including a C-terminal pHAT domain, structural studies of the full-length enzyme have been hampered by the presence of several predicted large disordered regions (Fig. 1a). Indeed, in three published crystal structures of the homodimeric hOGA catalytic core, these regions and the pHAT domains were not included in the expression constructs[17–19]. To facilitate an understanding of the structure and function of the pHAT domain, we turned to *T. adhaerens*, the simplest metazoan encoding for a complete O-GlcNAc system, but with a simpler architecture (Fig. 1a)[10]. As a first step towards a full-length *Ta*OGA structure, we determined the crystal structure of its pHAT domain (*Ta*pHAT). Protein purified from *E. coli* produced crystals suitable for structure determination at 1.8 Å (Fig. 1b and Supplementary Table 1). The *Ta*pHAT structure reveals two molecules in the asymmetric unit, with their β1-strands packing in a parallel fashion (Fig. 1b and Supplementary Fig. 1b). Consistent with the predicted pHAT canonical GCN5 acetyltransferase fold[34], the

overall topology of *Ta*pHAT consists of a central core of seven anti-parallel β-sheets surrounded by six α-helices (Fig. 1b). A 995 Å² inter-domain interface surface area, calculated with PISA[35], together with the *Ta*pHAT size-exclusion chromatography elution profiles and cross-linking experiments (Supplementary Fig. 1a) suggest that the dimer observed *in crystallo* is also present in solution. Additionally, alanine substitution of individual residues within this predicted interaction network neither disrupted *Ta*pHAT dimer formation, nor affected the thermal stability of the *Ta*pHAT motif, as measured by differential scanning fluorometry (DSF) (Supplementary Fig. 1d). These results suggest that the β1 strand is the driver behind the dimerisation of the isolated *Ta*pHAT and human pHAT domains (Supplementary Fig. 1a).

To assess whether this pHAT homodimeric arrangement is observed among other GCN5 family members, we used the PISA server[35] to find structures at least 70% similar to the observed *Ta*pHAT dimeric interface, but none were found, indicating that this is a unique arrangement. We searched for proteins with a fold similar to the *Ta*pHAT monomer, regardless of their oligomeric arrangement. Among the hits, only two structures had a *Q* score (1 for identical, 0 for dissimilar proteins) higher than 0.5: The *Oceanicola granulosus* bacterial HAT (*Og*HAT) (PDB 3ZJ0[24], *Q* score 0.62) and the *Saccharomyces cerevisiae* pseudo-HAT domain (*Ss*pHAT) (PDB 4BMH[36], *Q* score 0.54). None of these are known to form dimers. Structural alignment showed that the monomeric *Og*HAT and *Ss*pHAT possess a shorter β1-strand compared to *Ta*pHAT, possibly suggesting this as the underlying cause of the observed differences in oligomerisation (Supplementary Fig. 1c). The closest hit reported to form a biological dimer is the histone acetyltransferase protein from the archaeon *Pyrococcus horikoshii* (*Ph*HAT) (PDB 1WWZ[37], *Q* score 0.42). However, inspection of the structural alignment between *Ta*pHAT and *Ph*HAT reveals that the β1-strand of *Ph*HAT is shorter and not conserved, resulting in a dissimilar homodimeric interface (Supplementary Fig. 1c). In the *Ta*pHAT, the dimer interface is mostly formed by the parallel packing of the β1 strands plus the interaction of the α5 helix from one monomer with the middle part of the β1 strand from the other monomer. In contrast, the *Ph*HAT dimeric interface is exclusively formed by α-helical interactions, and, under this arrangement, the β1 strands are pushed away from the dimerisation interface (Supplementary Fig. 1c).

It should be noted that the human pHAT domain was not included in the PISA structural comparison due to the absence of a high-resolution experimental structure in the Protein Data Bank (PDB). Consequently, its potential to adapt a dimeric interface analogous to *Ta*pHAT could not be evaluated through this analysis. To address this, we investigated whether the *Ta*pHAT dimer interface is also present in the human pHAT. Similar to the *Ta*pHAT, the isolated human pHAT exhibited a size-exclusion chromatography elution profile indicative of a dimer (Supplementary Fig. 1a). Sequence inspection reveals that, despite having a mostly conserved β1 strand amino acid composition (40% sequence identity), the α5 helix composition is not conserved (Supplementary Fig. 3). This difference likely leads to loss of the network of electrostatic interactions in the putative human pHAT dimer (Supplementary Figs. 1c, d). Interestingly, although the isolated human pHAT domain forms a dimer, it has a melting temperature 12 °C lower than the *Ta*pHAT domain, perhaps explaining why the human pHAT has failed in crystallisation trials (Supplementary Fig. 1d). Taken together, the results indicate that, when expressed in isolation, the *Ta*pHAT and human pHAT motifs form homodimers, though possible differences in their dimer interface may impact their stability.

### Eukaryotic pHAT domains are unable to bind acetyl-CoA, but retain an acceptor binding cleft

Prior studies reported that the hOGA pHAT domain is unable to bind acetyl-CoA[24]. To investigate potential species-specific differences, we used surface plasmon resonance to analyse the acetyl-CoA binding properties of OGA pHAT domains from a range of organisms across

evolution (Fig. 1c). Consistent with earlier reports, none of the eukaryotic pHAT domains tested, including the *Ta*pHAT domain, showed acetyl-CoA binding, while the *Og*HAT[24], used here as a positive control, bound acetyl-CoA with a $K_d$ = 5 mM (Fig. 1c). To understand the lack of acetyl-CoA binding, we took advantage of the *Tetrahymena* HAT domain (tGCN5) structure (PDB 1Q2D[38]) crystallised in a ternary complex with acetyl-CoA and a 19-residue p53-derived acceptor peptide bound in a conserved cleft[38]. Inspection of the superimposed *Ta*pHAT with tGCN5 (RMSD of 4.8 Å across 80 Cα atoms), reveals steric clashes of the acetyl-CoA with the *Ta*pHAT p-loop, which is crucial in the GCN5 family members for stabilising the negatively charged acetyl-CoA pyrophosphate (Fig. 1d)[34]. Furthermore, the presence of four additional residues in the *Ta*pHAT α5 helix is likely to obstruct acetyl-CoA binding (Fig. 1d and Supplementary Fig. 1e). This architecture is conserved in the human pHAT domain (Supplementary Fig. 1e).

Although acetyl-CoA binding to the pHAT domains may be affected, a comparison of *Ta*pHAT and tGCN5 structures reveals the presence of a partially conserved peptide acceptor binding cleft (Fig. 1d, e). This cleft, formed by the α1 and the α6 helices, is blocked on one side by the presence of additional α3 and α4 helices in the *Ta*pHAT domains, which are not present in the canonical GCN5 fold (Fig. 1e)[39]. A deeper analysis using the DALI server[40] identified the cryo-EM structure of a macromolecular complex (PDB 7VVU[41]), between the yeast nucleosome and a histone acetyltransferase (NuA4) complex, where the HAT domain is structurally similar to the *Ta*pHAT domain (RMSD of 3.8 Å across 96 Cα atoms). An isolated view of this macromolecular complex shows the NuA4 HAT engaging the histone 4 (H4) tail, which penetrates one side of the NuA4 HAT cleft and is poised for acetylation, while the Polycomb complex-related protein Epl1 interacts on the opposite side, stabilising NuA4 binding to the nucleosome (Supplementary Fig. 1f)[41]. The superposition of the *Ta*pHAT domain structure with this complex reveals that the extra element formed by the α3 and α4 helices clashes with the H4 tail similarly to the clashes observed with the tGCN5 peptide substrate (Fig. 1e and Supplementary Fig. 1f). However, there are no apparent barriers for a putative interaction of the pHAT domain with the Polycomb related protein Epl1 (or any other nuclear protein) via the pHAT conserved binding cleft (Fig. 1e and Supplementary Fig. 1g). Taken together, these data suggest that while the pHAT domains of OGAs cannot bind acetyl-CoA and therefore are catalytically inert, it is likely that they retain the ability to interact with other proteins via the remnants of a GCN5 family acceptor binding cleft.

## *Ta*OGA possesses a conserved homodimer interface with flexible pHAT domains

To determine the spatial arrangement of the pHAT domains relative to the catalytic core, the full-length *Ta*OGA structure was solved by cryo-EM. A sample purified from *E. coli* was vitrified on a grid and imaged, with data processed using CryoSPARC (Supplementary Fig. 2)[42]. Particles were extracted using a 480 px box size to capture both the catalytic core and pHAT domains. Initial 2D classification revealed a well-defined homodimeric catalytic core but lacked clear density for the pHAT domains (Fig. 2a). Subsequent 3D reconstruction produced a map resolving the *Ta*OGA homodimeric catalytic and stalk domains at 3.0 Å resolution, into which an AlphaFold3[43] model guided model building (Fig. 2b).

The dimerisation interface of *Ta*OGA buries 2700 Å² of the total 20,000 Å² surface area, accounting for 14%, as calculated by the PISA server[35], which is 20% smaller than the reported value for the hOGA crystal structures[18,19]. The interface is largely conserved, differing at only a few residues, and forms a broad solvent-exposed hollow channel through the homodimer. Gln234 (Gln288 in hOGA) from each stalk domain extends into this channel, partially narrowing it to an inter-Cα distance of 7 Å in *Ta*OGA and 8 Å in hOGA (Fig. 2c and Supplementary Fig. 3a). The helical stalk region preceding the linker to the pHAT

domain further stabilises the dimer by closing the channel between the monomers, while the downstream linkers remain undefined and do not interact with the stalk domains. Notably, the interface is looser between the catalytic domains at the top and tightens near the active site, suggesting a balance between conformational flexibility and rigidity required for precise substrate positioning. The top of the stalk domains of this *Ta*OGA cryo-EM structure are in the same tight arrangement as in the hOGA catalytic core crystal structures, whereas one of the *Ta*OGA catalytic domains rotates away from the neighbouring stalk domain, which is reflected in a pairwise inter-Cα distance of 33 Å in *Ta*OGA (Lys59) and 40 Å in hOGA (Val113) (Fig. 2c).

The substrate binding residues in the active site are conserved and align with the hOGA crystal structure (PDB 5VVO[19]), except for a displacement of the loop containing the catalytic acid Asp121 (Asp175 in hOGA) (Fig. 2e, f). In the hOGA crystal structure, this loop has a small helical region interacting with the surface of the catalytic core, whereas this region is an unstructured displaced loop showing poor density in the *Ta*OGA cryo-EM structure (Fig. 2d–f). A break in the loop density places Asp123 in the GlcNAc binding site, while non-conserved residues Arg169, Gln431, His434, and Tyr464 (corresponding to Phe223, Gly619, Met622, and Ser652 in hOGA) (Supplementary Fig. 3a) coordinate a large unassigned additional density (Fig. 2e) at the entrance to the active site coordinated by Arg169, Tyr464 and His434. Taken together, these interactions provide insight into the conserved active site architecture and a non-conserved entry pathway, suggesting potential regulatory mechanisms of *Ta*OGA.

The absence of visible pHAT domains in 2D classifications and 3D reconstruction suggests substantial flexibility between the catalytic core and pHAT domains, despite the presence of a short linker (STDEYEESTLKNS) connecting them (Figs. 1a, 2a, b, and Supplementary Fig. 3a). This is consistent with AlphaFold3 predictions, which assign low pLDDT values to the linker region, indicative of low confidence in structural ordering and suggestive of local flexibility that could decouple the pHAT domains from the catalytic core (Supplementary Fig. 3b). Notably, several negative charges in the linker may contribute to electrostatic repulsion of these regions in the homodimer. However, given that both the pHAT and catalytic/stalk domains are symmetric homodimers—and AlphaFold3 similarly predicts a symmetric multi-domain homodimer, the two-fold axes in the *Ta*OGA pHAT homodimer crystal structure and the *Ta*OGA homodimer cryo-EM structure was aligned to predict the position of the pHAT domains relative to the catalytic cores (Fig. 2g). In this model, the pHAT domains are positioned between the stalk domains but do not interact with them. This results in the putative pHAT peptide binding sites pointing away from steric hindrance, facing the glycoside hydrolase active sites. Nevertheless, cryo-EM data suggest potential flexibility in the arrangement of the pHAT domains relative to each other and/or the catalytic core dimer. Taken together, these data suggest that *Ta*OGA, like hOGA, is a homodimer with dimerisation interfaces for the catalytic cores and pHAT domains, although the latter are averaged out due to the flexible linkers under conditions of the cryo-EM experiment.

## hOGA cryo-EM structures suggest discretely disordered pHAT domains

The role of the pHAT domain in regulating hOGA catalytic activity remains poorly understood. To investigate this, a recently published stem cell system in which endogenous OGT and OGA are fluorescently tagged was used, allowing quantitative assessment of how exogenously expressed OGA or OGT (or variants thereof) influence O-GlcNAc homoeostasis (Supplementary Fig. 4a)[44,45]. In this assay, expression of a pHAT-less hOGA variant (OGA-Cata) caused a significant disruption of O-GlcNAc homoeostasis compared to a full-length hOGA (OGA-WT). This was characterised by a larger increase in endogenous OGT levels and a larger decrease in both endogenous OGA and global O-GlcNAc

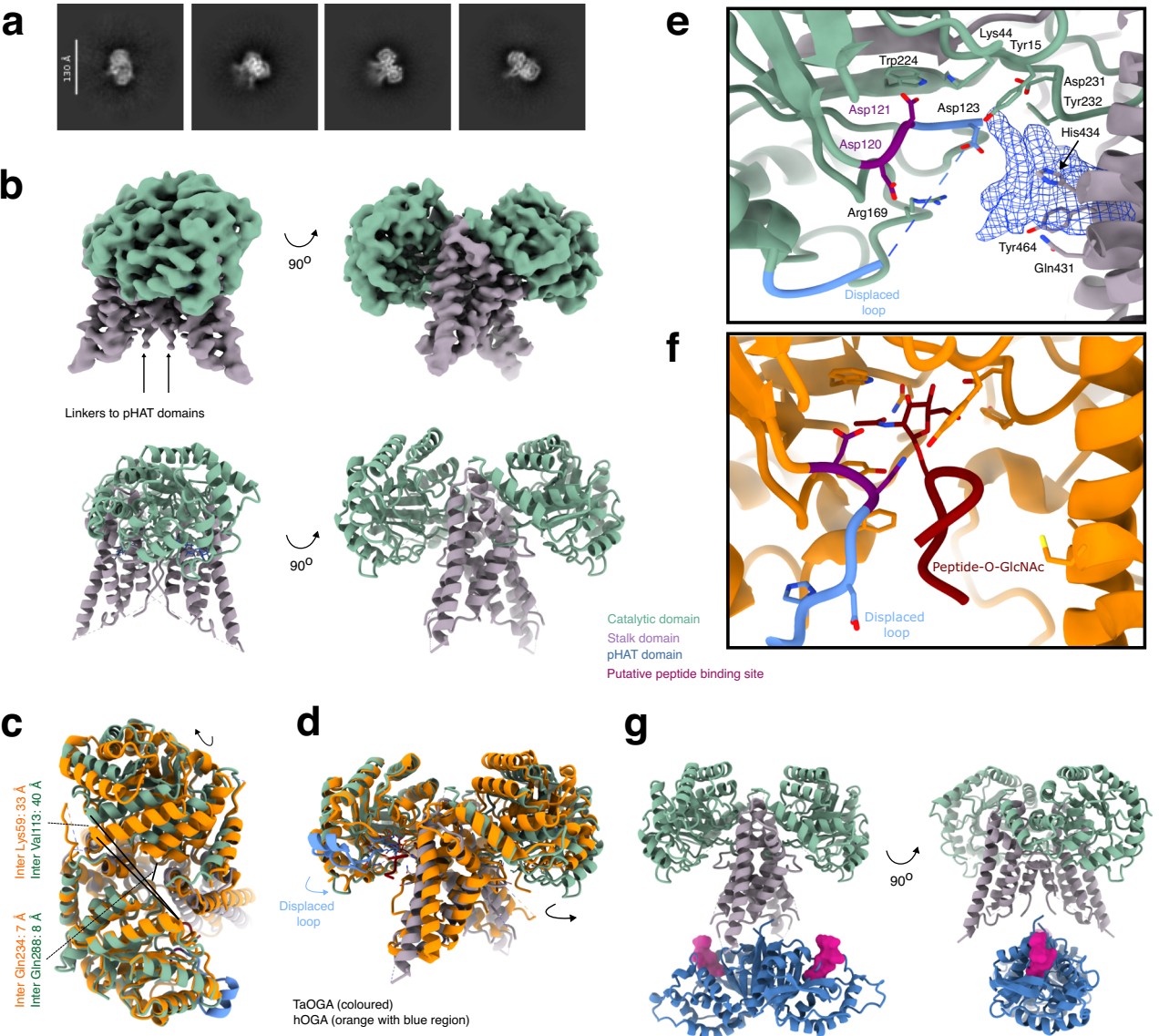

**Fig. 2 | Cryo-EM structural analysis of *Ta*OGA. a** Two-dimensional classes reveal the catalytic core, but the pHAT domains are not visible. **b** Particles were reconstructed to 3 Å applying two-fold symmetry, without density for the pHAT domains and a density break in the linker region, indicating flexibility of the pHAT domains. **c, d** Superposition of *Ta*OGA and hOGA with a glycopeptide bound in the active site (PDB 5UN8[17]) (orange) shows a conserved dimer interface with a tightly connected stalk domain and a looser arrangement between the catalytic domains (**c**) and a displaced loop region in blue (**d**). Arrows indicate displacement of the catalytic domain of *Ta*OGA relative to the hOGA crystal structure. Lines indicate pairwise inter-Cα distances. **e** Zoom-in of the conserved active site, where the displaced loop occupies the GlcNAc binding site, and a non-conserved putative binding site for a yet unidentified compound. The Asp-Asp catalytic dyad is coloured magenta. **f** Zoom-in of the active site of hOGA with a glycopeptide. The Asp-Asp catalytic dyad is coloured magenta. **g** Hybrid model of multi-domain *Ta*OGA using the crystal structure of the *Ta*pHAT domain dimer, the *Ta*OGA cryo-EM structure guided by an AlphaFold3 model, also showing the equivalent position of the P53 peptide obtained from a *Tetrahymena* HAT (PDB 1Q2D[38]) structure to indicate the position of the pHAT domain putative acceptor peptide binding site (pink). The *Ta*OGA model and map are coloured according to the the scheme shown in Fig. 1: catalytic domains (green), stalk domains (purple), and pHAT domains (blue).

levels (Supplementary Fig. 4b-d; adjusted $p < 0.0001$ for OGT-sfGFP and mScarlet3-OGA levels, and adjusted $p = 0.0086$ for O-GlcNAc levels; ordinary one-way ANOVA). These results are consistent with previous findings[18] and suggest that removal of the pHAT domains enhances hOGA activity, potentially by altering the OGA interactome to broaden substrate selectivity or by increasing catalytic turnover due to greater accessibility of the active site in the absence of the pHAT domains. Together, these observations support a role for the pHAT domain in allosteric regulation of hOGA. Overexpression of the pHAT domain alone (OGA-HAT) had no impact on O-GlcNAc homoeostasis, as endogenous OGT and OGA levels, as well as global O-GlcNAc levels, remained unchanged (Supplementary Fig. 4b-d; adjusted $p > 0.9999$ for OGT-sfGFP, mScarlet3-OGA, and O-GlcNAc levels between OGA-

HAT and Mock transfection; ordinary one-way ANOVA). Similarly, overexpression of catalytic inactive pHAT-less hOGA variant (OGA-Cata(D175N)) had no significant effects on O-GlcNAc homoeostasis.

To further investigate the allosteric relationship between the catalytic core and pHAT domains of hOGA, we determined its cryo-EM structure. A construct was designed that lacks the large, disordered loop (residues 397–535, Fig. 1a) while retaining the pHAT domains. As with *Ta*OGA, this hOGA construct was expressed and purified from *E. coli*, imaged by cryo-EM, and processed using CryoSPARC (Supplementary Fig. 5)[42]. Both 2D and 3D classifications showed particle classes with well-defined catalytic and stalk domains, but none of these included fully defined pHAT domains (Fig. 3a, upper panels, b). Instead, these domains appeared as blurred density in 2D and as an

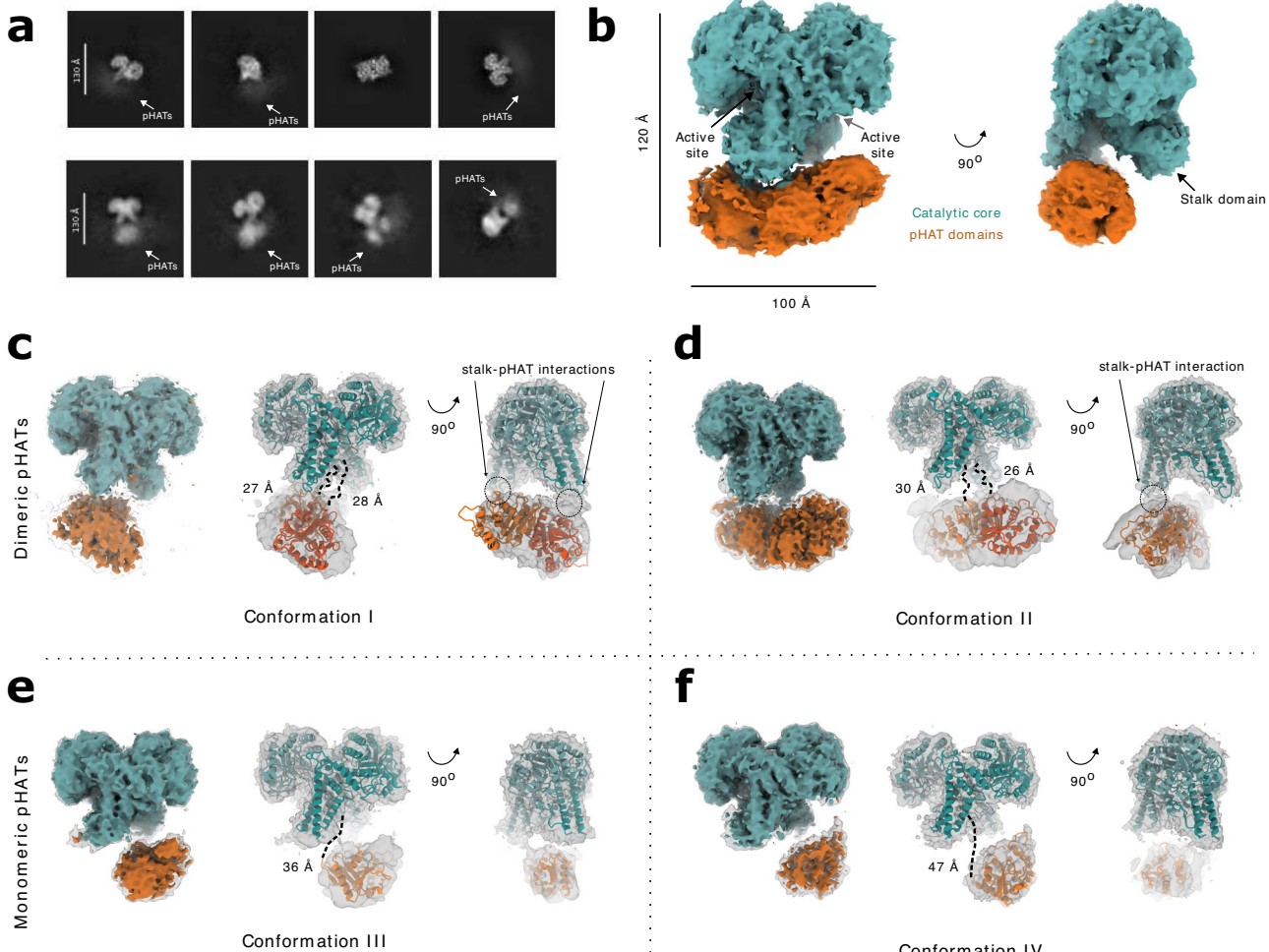

**Fig. 3 | Conformational variability of the hOGA pHAT domains. a** Two-dimensional classes with 5–6000 particles reveal a well-ordered catalytic core, and blurred density indicates positional heterogeneity for the pHAT domains (upper panel). Two-dimensional classes with 1000 particles reveal pHAT domains in various positions (lower panel). **b** 140,000 particles were reconstructed to 3 Å. The catalytic core is well-defined, but the pHAT domains appear as an elongated density with multiple positions. Subclassification resolved four different conformations (I – IV) of the pHAT domains. **c, d** In conformations I (**c**) and II (**d**), the pHAT domains interact as dimers, where the pHAT domains interact with the extended loop of the stalk domains. **e, f** In conformation III (**e**) and IV (**f**), the pHAT domain is found as two separate monomers released from interacting with the stalk domains, with only a single pHAT domain being resolved. The distance between the final ordered residue in a catalytic core (Leu692) and the first resolved residue of the pHAT domain (Lys713) is measured in (**c–f**). The catalytic core is coloured in teal and the pHAT domains are orange.

elongated asymmetric density between the two stalk domains in 3D, suggesting that the pHAT domains adopt a range of different angular positions relative to the catalytic core. Sorting the particle stack into less populated 2D classes showed visible pHAT domains in various positions, suggesting pHAT domain positional heterogeneity (Fig. 3a, lower panels). This variation suggests that certain conformations are more frequently sampled, potentially reflecting energetically preferred states within the broader conformational landscape. To further explore this heterogeneity, particles were sub-classified to enrich for more populated conformations. This analysis yielded four major conformations—termed I, II, III and IV—each separately subjected to 3D reconstruction and refined to global resolutions ranging from 3.3 Å to 4.0 Å (Fig. 3c–f). By focusing on these dominant conformations, the analysis introduced a bias towards the most stable or frequently sampled states, effectively masking lower-populated, potentially transient intermediates. An AlphaFold3 model of hOGA was used to guide model building, and across all conformations the catalytic core consistently displayed a near two-fold symmetry, forming two glycoside hydrolase active sites composed of the catalytic domain of one monomer and the stalk domain of the other (Fig. 3b). Root-mean-

square deviation (RMSD) values of 0.7–0.8 Å confirm that the catalytic core closely resembles that of the published hOGA catalytic core crystal structure (PDB 5VVO[19]).

While the catalytic core was well-defined, the pHAT domains were not consistently defined in the maps, likely due to their conformational variability. To approximate the positions of the pHAT domains, a blurred density map was generated, providing a coarse view of their locations (Fig. 3c–f). A composite map was constructed by combining the high-resolution density of the catalytic core with the blurred density of the pHAT domains, offering an overall structural context. The pHAT domains were subsequently docked into the density using ChimeraX[46], guided by either a dimeric or monomeric hOGA pHAT model predicted by AlphaFold3.

The four conformations identified represent distinct arrangements of the pHAT domains relative to the catalytic core, highlighting the structural flexibility of hOGA. In conformation I, the pHAT domains form a dimer that interacts with an extended loop region in the stalk domains (residues 594–600, Supplementary Fig. 3a and Fig. 4c). In this state, the density indicates that the dimeric pHAT interacts with the stalk domains, although residual flexibility is implied by the density.

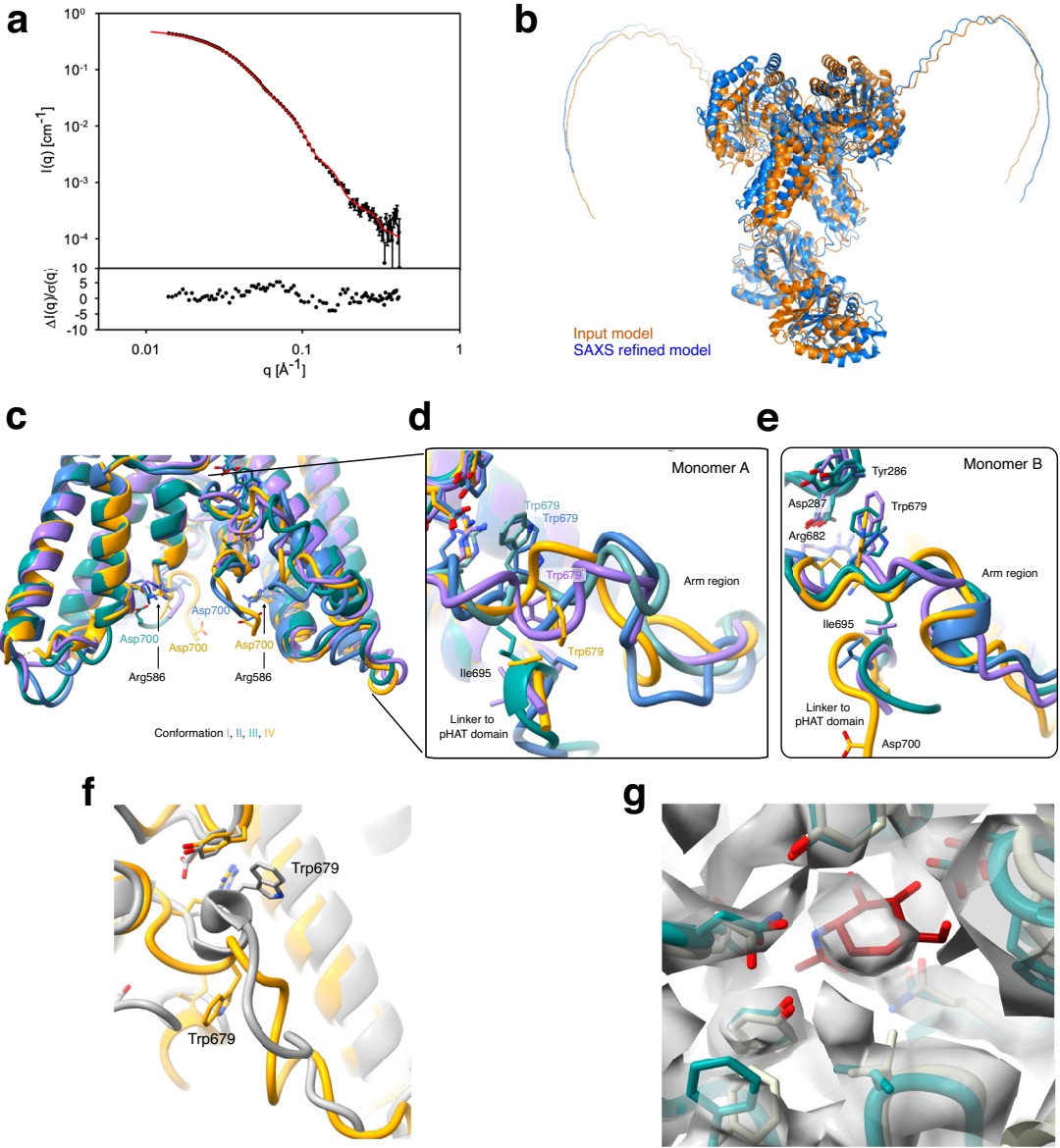

**Fig. 4 | SAXS solution shape of hOGA and pHAT domain-induced conformational changes. a** SAXS data and fit from the best rigid-body refinement model. **b** Comparison of the best hOGA SAXS fitted model (blue) with the hOGA input model (orange) aligned on the stalk region. **c–e** Structural comparisons of conformations I (purple), II (blue), III (teal) and IV (yellow), covering the stalk domains with visible linkers to the pHAT domains (**c**), and the arm region from both monomer A (**d**) and monomer B (**e**). **f** Structural comparison of the arm region of conformation IV and hOGA crystal structure (grey) (PDB 5VVO[19]). **g** Close-up of the active site showing an averaged map of particles from conformation I–IV (teal), with extra density likely corresponding to a GlcNAc molecule (red sticks).

Mutating two lysines (Lys597Ala and Lys599Ala; OGA-2KA) in this loop region in an attempt to decouple this stalk-pHAT interaction does not alter hOGA activity, indicating that this interaction is not critical for regulating the activity of hOGA or substrate selection (Supplementary Fig. 4b-d). In conformation II, the pHAT domains retain a dimeric configuration but with only a single pHAT domain interacting with one of the stalk domains (Fig. 3d). In this state, the other pHAT domain is stabilised solely through its dimerisation interface, while its connection to the stalk is not defined. Confirmation III reveals a monomeric pHAT domain interacting with a single stalk domain, while the other pHAT domain is unresolved (Fig. 3e). The linker region connecting the catalytic core to the pHAT domain exhibits some density, indicating partial stabilisation. Indeed, mutation of two key phenylalanines in the linker (Phe703Ala and Phe704Gly; OGA-2FAG) leads to a milder disruption of O-GlcNAc homoeostasis, as transfection with the OGA-2FAG

construct still perturbed O-GlcNAc homoeostasis, albeit significantly less than OGA-WT transfection (Supplementary Fig. 4b-d; adjusted $p < 0.0001$ for OGT-sfGFP and mScarlet3-OGA levels, as well as O-GlcNAc levels, between OGA-2FAG and OGA-WT transfection, ordinary one-way ANOVA). This highlights the importance of the composition of the linker to retain OGA stability. Conformation IV, similar to conformation III, also features a single monomeric pHAT domain interacting with a stalk domain (Fig. 3f). However, in this state, the pHAT domain is rotated closer to the catalytic core, and the density for the linker region disappears, perhaps reflecting a loss of ordered interactions with the stalk domain.

A comparative analysis of these conformations reveals several trends. Conformations I and II, featuring dimeric pHAT domains, exhibit stronger interactions with the stalk domains, particularly at their loop regions. In contrast, conformations III and IV, characterised

by monomeric pHAT domains, exhibit better resolved density for the pHAT domains and reduced variability in their positions. This suggests that dimeric pHAT domains, when stabilised by the stalk domains, adopt more variable conformations, while monomeric domains are more ordered, at least one of them. The four conformations likely represent the four most stable states within a broad conformational landscape of hOGA. Such structural plasticity may be crucial for the function of hOGA, enabling it to interact with a diverse array of substrates or regulatory partners.

### Multi-domain hOGA adopts asymmetric conformations in solution

To further explore the conformational heterogeneity in the hOGA dimer, we conducted a SAXS experiment using the same construct as for cryo-EM. The intensity $I(q)$ vs the scattering vector modulus ($q$) plots for hOGA are shown in Supplementary Fig. 6a and further used for a Guinier analysis. The fit of the Guinier residual plot ($\ln(I(q))$ vs $q^2$) is acceptable, suggesting that most of the sample is properly folded in solution with little aggregation. The zero-angle scattering intensity values of $I(0)$ and the radius of gyration $R_g$ are given in Supplementary Table 2. These values were used to generate the dimensionless Kratky plots (($R_g q$)$^2$ $I(q)/I(0)$ vs ($R_g q$)) (Supplementary Fig. 6b). After the main broad bump, which originates from the folded domains, the data reach a plateau at high $q$, indicating the presence of some flexible parts or loops in the hOGA structure. To gain further insights into the size and shape of the proteins, the pair distance distribution function $p(r)$ was generated using either the GNOM program of the ATSAS package 4.0[47] or BIFT[48] using the server at https://somo.chem.utk.edu/bayesapp/ (Supplementary Fig. 6c). Both the GNOM and BIFT $p(r)$ functions suggest a compact structure with a maximum diameter of 190–200 Å for hOGA. The radius of gyration determined by GNOM and BIFT aligns well with that from the Guinier fit, indicating consistency between these methods.

To assess the agreement between the cryo-EM map of hOGA and the corresponding SAXS data, a recently published program[49] was used to fit these data together by calculating the SAXS intensity from the density map. Although the agreement between computed SAXS profiles using the cryo-EM map and the experimental SAXS data is not perfect (Supplementary Fig. 6d), the comparison shows that the cryo-EM map reasonably well represents the structures in solution, accounting for the unobserved flexible and disordered regions. To obtain additional structural insights, SAXS data were modelled using high-resolution models obtained from the hOGA catalytic core crystal structure[17,18] and a human pHAT 3D model derived from the *Ta*pHAT crystallographic structure via Modeller[50]. These models were further optimised by rigid-body refinement. The resulting models had reduced chi-square values, $\chi^2$, of 2.9–6.5 for an average oligomerisation of the dimers with ~60–70% dimers of dimers with an inter-dimer centre-of-mass distance of 60–70 Å. These aggregated dimers are also supported by the masses derived from the Porod and the correlation volumes, which are about 280 kDa compared to 169 kDa for the dimer itself (Supplementary Table 2). The dimer structure that gives the best fit to the SAXS data is shown together with the input model, which has close to C2 symmetry (Fig. 4a, b). In this best-fit SAXS model, the pHAT domains maintain contact with the stalk domains but are loosely connected to the rest of the structure. These results align SAXS data with the observed flexibility in the hOGA cryo-EM data, corroborating the conclusion that the pHAT domains in hOGA adopt multiple conformations in solution, breaking the expected C2 symmetry and generating asymmetric dimers.

### The pHAT domain and its linker allosterically regulate hOGA

In all four conformations of hOGA (Fig. 3c–f), the cryo-EM density maps reveal a discontinuity in the region following Leu701, corresponding to the linker connecting the catalytic core and the pHAT domains. This linker (LFFQPPPLTPTSKVY, Supplementary Fig. 3a) is of medium length and proline-rich, characteristics known to disrupt secondary structures and enhance conformational flexibility[51,52]. Such intrinsic disorder likely contributes to the observed positional heterogeneity of the pHAT domains. The spatial separation between the final ordered residue in a catalytic core (Leu692) and the first resolved residue of the pHAT domain (Lys713) varies depending on the orientation of the domains: In conformations I and II, where the pHAT domains engage with the extended loop of the stalk domain, the distance ranges from 26 Å to 28 Å (Fig. 3c, d); in conformation II, where this interaction is lost, it extends to 30 Å (Fig. 3d); and in conformations III and IV, where the pHAT domains are released from stalk domain, the linker stretches further, reaching 36 Å and 47 Å, respectively (Fig. 3e, f). These distance variations suggest that interactions with the stalk domain stabilise the pHAT domains, while disruption of these interactions allows increased conformational mobility, mediated by the flexible linker. This dynamic positioning of the pHAT domains may have functional implications for hOGA regulation.

Crystal structures of the hOGA catalytic core have previously left the arm region and the extended linker within the stalk domain largely unresolved. In contrast, our hOGA cryo-EM structures containing the pHAT domains reveal defined, albeit at lower resolution, density for both regions (Supplementary Fig. 5), suggesting that the presence of the pHAT domains stabilises these otherwise flexible regions. Like previous crystal structures[17–19], the arm region appears to adopt distinct conformations predominantly in one monomer at a time, potentially reflecting an induced-fit mechanism, in which one protomer samples alternative states while the other remains comparatively stable.

The linkers to the pHAT domains interact with key amino acids on the inner surface of the stalk domains, anchoring the arm region in various conformations (Fig. 4c). The position of the linker, and thereby its interaction with the arm region, modulates the exposure of residues lining the path to the active site (Fig. 4d,e). The arm region can undergo a conformational twist, with Trp679 serving as a reference point for this movement, alternating between orientations towards the linker or the catalytic domain (Fig. 4d). In this way, the position of the pHAT domain could allosterically regulate the exposure of amino acids in the environment leading to the active site, in agreement with the increase in O-GlcNAcase activity seen upon loss of the pHAT domains (Supplementary Fig. 4b-d).

Due to the moderate resolution in these regions, precise interactions are difficult to resolve, but in conformation IV, the linker is brought into proximity of several residues within the stalk domain: Ile695 seems to interact with Trp679 in the arm region, while Asp700 appears positioned to form a salt bridge with Arg586 (Fig. 4e). In previously published crystal structures of the catalytic core only, where the arm region is partially defined, Trp679 orients towards the catalytic domain (Fig. 4f), suggesting that the arm region may adopt different conformations depending on linker positioning. To probe the functional relevance of these potential interactions, we tested mutations in linker residues identified to contact the inner surface of the stalk domain (Pro694Gly, Ile695Ser, and Asp700Ala; OGA-PGISDA). In the cellular assay, transfection with the OGA-PGISDA construct was found to disrupt O-GlcNAc homoeostasis, albeit to a significantly lesser extent compared to the OGA-WT construct (Supplementary Fig. 4b-d; adjusted $p < 0.0001$ for OGT-sfGFP and O-GlcNAc levels, adjusted $p = 0.0007$ for mScarlet3-OGA levels; ordinary one-way ANOVA). This shows that the linker composition is essential to maintain functional O-GlcNAc homeostasis. In contrast, mutating Trp679Ala had no apparent effect (Supplementary Fig. 4b-d), indicating that while it may serve as a useful reference point for arm positioning, it is not essential for catalytic activity or substrate selection.

Interestingly, the cooperativity between the pHAT and stalk domain may also extend into the active site. Unlike the crystal

structures of hOGA without pHAT domains, all four conformations display additional density in the active site (Fig. 4g). To identify the bound molecule and enhance the signal, all particles were aligned. Superimposing a crystal structure of hOGA with GlcNAc in the active site suggests a density compatible with GlcNAc, presumably co-purified with the enzyme from *E. coli* lysates.

To investigate whether the observed pHAT-mediated regulation is conserved across species, we replaced the human linker with that from *Ta*OGA (OGA-linker-Ta) and tested it in the cellular assay. Unlike transfection with OGA-WT, this chimaera did not disrupt O-GlcNAc homoeostasis to the same extent (Supplementary Fig. 4; adjusted $p < 0.0001$ for OGT-sfGFP, mScarlet3-OGA, and O-GlcNAc levels between OGA-WT and OGA-linker-Ta transfection, ordinary one-way ANOVA), and a similar result was observed with a chimaera of equal linker length (OGA-linker-Ta + 5; adjusted $p < 0.0001$ for OGT-sfGFP, mScarlet3-OGA, and O-GlcNAc levels between OGA-WT and OGA-linker-Ta + 5 transfection, ordinary one-way ANOVA). These results underscore the central role of the linker in allosteric regulation and reflect evolutionary divergence in OGA regulation between humans and *T. adhaerens*.

## Discussion

All proteins capable of hydrolysing O-GlcNAc belong to the GH84 family, which often couple O-GlcNAcase activity with other functions, possibly to facilitate proper localisation or regulate activity. This functional diversity arises from the fusion of the GH84 catalytic core with additional domains. In bacteria, GH84 proteins frequently associate with domains mediating protein–protein interactions, such as coagulant factors, fibronectin type III domains or cohesion and the dockering domains[53–55]. In metazoans, the GH84 catalytic core exists both as a single domain protein or fused to C-terminal pHAT domains[10,56], suggesting functional interdependence. The evolutionary link between the pHAT and GH84 catalytic domains raises intriguing questions, since disordered linkers in multi-domain proteins often function to increase the local concentration of interacting domains, thereby facilitating allosteric regulation[33].

In this work, we demonstrate that deletion of the pHAT domain induces a more pronounced disruption in O-GlcNAc feedback homoeostasis, suggesting increased catalytic activity or broader substrate promiscuity (Supplementary Fig. 4b-d). These findings highlight the intricate regulation of hOGA, in which conserved binding pockets of the pHAT domains are positioned facing the catalytic core active sites (Fig. 3g). This arrangement likely provides a fine-tuned mechanism for modulating activity, possibly by influencing a broad substrate recognition through creation of a stable, yet dynamic, peptide binding cleft, mediated by interactions between the arm region in the stalk domains and catalytic core-pHAT linkers (Fig. 4b-d). To further investigate this interaction, we mutated different amino acids in these linkers, which resulted in significantly reduced disruptions in O-GlcNAc homoeostasis compared to that from wild type OGA (Supplementary Fig. 4b-d). This suggests that the interaction between this linker region and the stalk domain is essential for activity. Disrupting this interaction could lead to a "pull" from the now decoupled pHAT domains, potentially rearranging the tightly organised dimerisation interface near the active site and thereby preventing catalysis. Overall, our results indicate that the pHAT domains and their linkers regulate hOGA activity through allosteric modulation, stabilising a dynamic peptide-binding cleft and maintaining the dimerisation interface, which ensures proper substrate recognition and O-GlcNAc homoeostasis.

The structural and functional consequences of this regulation are particularly intriguing in the context of the conformation flexibility of hOGA, given that multimeric proteins often depend on dynamic rearrangements to facilitate interactions with various partners[57,58]. In some systems, pronounced asymmetry within homodimers can be functionally significant by limiting the active site availability or

enabling interactions with ligands, such as DNA[59]. Within this framework, the functional role of these C-terminal HAT-like domains remains controversial. While initially proposed to have HAT activity[23], previous work[24,25], along with our data shown here, show that eukaryotic HAT-like domains cannot bind acetyl-CoA, classifying them as pHATs of unknown function. Notably, the pHAT domains retain a putative peptide binding groove (Fig. 1e), implicating them in specific protein–protein interactions rather than enzymatic activity. Indeed, these domains are critical for hOGA localisation to the nucleus following DNA damage[27], suggesting that their structural flexibility plays a key role in mediating the interactions between the putative peptide binding groove and unidentified chromatin components. The absence of catalytic function, coupled with the presence of this conserved peptide binding groove in the pHAT domain, supports that this domain has evolved to facilitate essential protein–protein interactions.

A recent preprint has also reported substantial flexible pHAT domains in hOGA, based on a single cryo-EM structure, validating our observations[60]. Additionally, evidence was provided that the pHAT domain could bind modified histone peptides, suggesting a role for the pHAT in engaging with nucleosome components. This interaction could position hOGA to modulate O-GlcNAcylation of components of the transcriptional machinery, including RNA Polymerase II.

The regulatory role of the pHAT domains in hOGA appears to have diverged from that in *T. adhaerens*, as revealed by differences in cryo-EM density for the pHAT domains (Figs. 3 and 2b), likely reflecting adaptations to their respective cellular environments. Notably, the linker connecting the catalytic core and the pHAT domain is smaller in *Ta*OGA and not conserved between the two organisms (Supplementary Fig. 3a), likely weakening the interaction between the catalytic core and the pHAT domains, resulting in no observed density for the pHAT domains in the cryo-EM structure of *Ta*OGA (Fig. 2b). This structural divergence suggests that *Ta*OGA and hOGA may have evolved distinct regulatory mechanisms suited to their functional requirements. As the *T. adhaerens* constructs did not express in our cellular system, it was not possible to measure the O-GlcNAc homoeostasis and compare it to the effect of the various constructs created for hOGA. Instead, we exchanged the human linker with the linker from *T. adhaerens* both with and without compensation for the difference in length, and the results from both constructs showed a significant change in O-GlcNAc homoeostasis compared to WT hOGA (Supplementary Fig. 4b-d), underscoring that the linker composition is essential for the allosteric regulation of hOGA. In humans, the greater complexity of O-GlcNAc cycling in cellular signalling, metabolism, and disease pathways may explain the additional regulatory intricacies suggested by a tighter association between the pHAT domains and the catalytic core of hOGA compared to the simpler multicellular *T. adhaerens*.

*OGT* missense variants leading to intellectual disability are associated with a compensatory loss of OGA mRNA and protein, leading to the intriguing possibility that the resulting impairment of pHAT domain function could also contribute to the disease mechanisms. Beyond its role in deglycosylation, targeting the pHAT domain or its interaction with the catalytic core could offer an approach for therapeutic intervention, since current inhibitors designed to competitively block the GH84 active site face limitations due to the broad range of OGA substrates. A more targeted approach—disrupting the pHAT-GH84 domain cooperativity—could yield greater specificity while preserving basal catalytic function.

Ultimately, hOGA is part of a complex regulatory network governing homeostasis of O-GlcNAcylation, a modification with widespread implications in cellular signalling, metabolism, and disease. Missense variants in hOGA, including the pHAT domain, are now beginning to be associated with neurodevelopmental disorders[8]. By elucidating the structural mechanisms underpinning hOGA regulation, we provide a platform for the dissection of the role of the pHAT

domain in mechanisms of modulating O-GlcNAc homoeostasis in such disease contexts and beyond.

# Methods

## Cloning, expression and purification of the recombinant pHAT domains

The pHAT region of *Ta*OGA (residues 512–723) was isolated from the whole gene[10,61] by PCR using the following primers: CTGGGATC-CACGTTAAAGAATTCTGATGCGTATG and GATGCGGCCGCTA-CATTAATTTAATAGCAATCATGC. The resulting PCR product was cloned as a *BamHI-NotI* fragment. Mutations in the *Ta*pHAT were introduced into the construct expressing GST-tagged *Ta*pHAT domain using site-directed mutagenesis based on the QuikChange site-directed mutagenesis kit from Stratagene. Primer pairs are listed in the Supplementary Table 3. Kod Hot Start Polymerase from Merck was used. Following PCR, the parent plasmid was digested using DpnI from Fermentas. Mutant plasmids were confirmed by DNA sequencing.

Mouse (710–916), chicken (705–911), zebrafish (701–909), *Drosophila* (792–1019) and human (710–916) pHAT domains were all obtained as codon-optimised gene blocks from Integrated DNA Technologies (https://eu.idtdna.com) bearing *BamHI* and *NotI* sites at their 5′ and 3′ ends, respectively. The sequences coding for the pHAT domains were inserted into a pGEX6P1 vector for expression of proteins with a GST-tag containing a PreScission protease cleavage site. The pGEX6P1 plasmid was subsequently modified to replace the GST-tag and PreScission cleavage site with a non-cleavable 6×His-tag.

The inserts from the pHAT domain clones were subcloned into this vector to express pHAT domains with fused 6×His-tags. For all the constructs, protein expression was performed as follows: First, the pHAT domain plasmids were transformed into *E. coli* BL21(DE3) pLysS. Cell cultures were grown to an $OD_{600}$ of 0.8, and then 300 μM IPTG was added to induce protein expression at 16 °C. The cultures were grown for an additional 16 h and harvested by centrifugation at 4000×$g$ in a J6-MI centrifuge (Beckman Coulter) equipped with the JS-4.2 rotor (Beckman Coulter). The cells were resuspended in lysis buffer (25 mM Tris buffer pH 7.5, 150 mM NaCl, and 0.5 mM TCEP containing DNase, protease inhibitor cocktail (1 mM benzamidine, 0.2 mM PMSF and 5 μM leupeptin) and lysozyme) and lysed using a French press (Thermo Scientific). Cell debris was pelleted by centrifugation at 21,000×$g$ in an Avanti J26S centrifuge (Beckman Coulter) equipped with the JA-25.50 rotor (Beckman Coulter). The GST-tagged pHAT supernatant was incubated with glutathione Sepharose beads for 2 h on a roller at 4 °C, while the His-tagged supernatant was incubated with nickel NiNTA beads. After intensive wash, the GST-tagged proteins were cleaved off the beads with Pre-Scission protease overnight at 4 °C, and the His-tagged samples were eluted from the beads with lysis buffer supplemented with 250 mM imidazole. Finally, all the proteins were concentrated and loaded onto a 26/600 Superdex 75 column (GE Healthcare) mounted in an AKTA system (GE AKTA Prime Plus) and equilibrated with lysis buffer. Fractions confirmed by SDS-PAGE were pooled and concentrated up to 30 mg/mL.

## Analytical size-exclusion chromatography

To analyse the oligomeric state of the pHATs, 500 μL of each His-tagged pHAT sample (*Ta*pHAT and hpHAT) at 1 mg/mL in (25 mM Tris buffer (pH 7.5), 150 mM NaCl, and 0.5 mM TCEP) was injected into a Superdex 200 Increase 10/300 GL column (Cytiva) mounted into an AKTA system (Cytiva AKTA go) equilibrated with the same buffer. Protein standards (Conalbumin, Ovalbumin, and Carbonic Anhydrase) from the LMW gel filtration calibration kit (Cytiva) were used as molecular weight references (MW range: 29–75 kDa) to estimate the molecular weight of the pHAT samples in solution.

## Crosslinking experiments

pHAT samples (0.5 mg/mL stock), dialysed into (50 mM HEPES-NaOH (pH 7.0), 150 mM NaCl), were incubated with glutaraldehyde at room temperature for 4 h. The reactions were then quenched by adding 10 μL of 1 M Tris-HCl (pH 8.5), which quenched the reactivity of glu-taraldehyde towards the protein amine groups. Reactions were pre-pared as described previously[62]. In brief, each sample contained a fixed amount of protein (9 μg) plus variable glutaraldehyde concentrations (0.0044 mM, 0.044 mM, 0.44 mM, and 4.4 mM) in a 20 μL final reaction volume. The glutaraldehyde stock (Thermo Scientific 25% ($v/v$) aq. soln) was diluted in HEPES buffer as needed. After incubation and quenching, the resulting crosslinked samples were resuspended in loading buffer, boiled for 5 min and then directly analysed by SDS-PAGE.

## DSF

pHAT DSF denaturing profiles were measured with a real-time PCR configured for detecting FRET signal (BioRAD CFX Opus 96). Briefly, 10 μM protein was mixed with 10 μM SYPRO Orange (Invitrogen ref S6650, 5× final working concentration) in (50 mM HEPES (pH 7.5), 150 mM NaCl). The final reaction volume was 25 μL. Experiments were performed in sextuplicate, and results were plotted and further analysed using GraphPad Prism.

## TapHAT domain crystallisation and structure solution

Ten milligrams per mL of pure tag-less *Ta*pHAT domain protein in (25 mM Tris (pH 7.5), 150 mM NaCl, and 0.5 mM TCEP) was used to screen for crystals at 20 °C using the sitting-drop vapour diffusion method. Several promising hits were identified in the INDEX 1&2 screens (Hampton). Following further manual optimisation, high-quality, diffracting crystals were obtained by mixing 0.6 μL of protein solution with an equal volume of a reservoir solution containing 0.1 M HEPES (pH 7.5), 60 mM sodium potassium tartrate, and 27.5% PEG 8000. The resulting crystals belonging to the P1 space group grew after 2 days. Before data collection, crystals were cryo-protected with 15% ($v/v$) glycerol in mother liquor, mounted in a nylon loop, and frozen in liquid nitrogen. X-ray datasets were collected at the ID30A beamline of the European Synchrotron Radiation Facility (ESRF, Grenoble, France). Data were processed with XDS[63]. The *Ta*OGA pHAT domain structure was solved using a *S. cerevisiae* HAT structure (PDB 4BMH[36]) as the initial phase donor in a molecular replacement experiment using MOLREP[64]. Refinement was performed with REFMAC5[65] and model building with COOT[66]. Data collection statistics are listed in Supplementary Table 1. The electron density maps covering the two molecules in the asymmetric unit are displayed in Supplementary Fig. 1b. Figures were generated using the PyMOL Molecular Graphics System, Version 2.5, Schrödinger, LLC.

## Cloning, expression and purification of hOGA and TaOGA

The hOGA protein was produced using a plasmid containing a 6×His-tagged N-terminal region of OGA (11–400) followed by the C-terminal region of the protein (535-end), bridged by a GS linker, as previously described by refs. 17,19. The insert was obtained as a codon-optimised optimized geneblock from Integrated DNA Technologies (https://eu.idtdna.com) bearing a *BamHI* and *NotI* at the 5′ and 3′ ends, respectively. The plasmid backbone pHEX6P1 is a modified version of pGEX6P1, but with a 6xHis-tag replacing the GST-tag. This vector adds a PreScission cleavable N-terminal 6×His-tag. The previously published full-length clone of *Ta*OGA was used as a template for PCR followed by restriction cloning. In addition to the previously used reverse primer, an additional primer was used of the following sequence— CTGGGATCCGATAAATTCCTTAGTGGCGTGG. The DNA region encoding residues 5–723 was cloned into pHEX6P1 as a *Bam*HI-*Not*I fragment. The resulting human and *Ta*OGA constructs were transformed into *E. coli* BL21(DE3) pLysS for protein expression. Cell

cultures were grown to an $OD_{600}$ of 0.6 and induced with 300 mM IPTG at 16 °C overnight. The cultures were harvested by centrifugation at 3500×$g$ for 20 min in a J6-MI centrifuge (Beckman Coulter) equipped with the JS-4.2 rotor (Beckman Coulter). Cells were resuspended in lysis buffer (25 mM Tris (pH 7.5), 150 mM NaCl and 0.5 mM TCEP) and supplemented with DNase, protease inhibitor cocktail (1 mM benzamidine, 0.2 mM PMSF and 5 μM leupeptin) and lysozyme, followed by lysis using a French press (Thermo Scientific). Cell debris was pelleted by centrifugation for 1 h at 21,000×$g$ in a refrigerated Avanti J26S centrifuge (Beckman Coulter) equipped with the JA-25.50 rotor (Beckman Coulter), and the supernatant was incubated with Ni-NTA beads for 2 h on a roller at 4 °C. Beads were extensively washed with lysis buffer, and bound proteins were eluted with 250 mM imidazole, followed by overnight incubation with PreScission protease to remove the 6×His-tag. Samples were concentrated and loaded onto a 16/600 Superdex 200 column equilibrated with (50 mM HEPES (pH 7.0), 250 mM NaCl and 0.5 mM TCEP). Fractions were confirmed by SDS-PAGE, pooled, concentrated to 10 mg/mL, flash frozen in liquid nitrogen and stored at −80 °C until further use.

### Surface plasmon resonance

SPR measurements were collected using a Biacore 3000 instrument (GE Healthcare). OGA pHAT domains were biotinylated by mixing the protein with amine-binding biotin (Pierce) in a 1:1 molar ratio. Streptavidin was immobilised on a CM5 sensor chip surface by amine coupling. 10 mM HEPES (pH 7.4), 150 mM NaCl, was used as a running buffer for immobilisation. The surface was activated by 15 min injection of NHS/EDC, followed by injection of SA in 10 mM acetate (pH 4.5) until the required density (approximately 9000 relative units (RU)) was achieved and blocked by 4-min ethanolamine injection at 10 μL/min at 25 °C. Biotinylated OGA pHAT domains were captured on the streptavidin surface at approximately 2500–4000 RU in running buffer containing 25 mM Tris (pH 7.5), 150 mM NaCl, 1 mM DTT and 0.005% Tween 20. Acetyl-CoA was injected in duplicates at threefold concentration series in a range of concentrations 0.2–166.6 μM. Association was measured for 30 s, and dissociation for 1 min. All experiments were run at 50 μL/min at 25 °C. All data were referenced for blocked streptavidin surface and blank injections of buffer. Scrubber 2 (BioLogic Software) was used to process and analyse data. Affinities were calculated using a 1:1 equilibrium-binding fit.

### Cryo-EM data collection and processing

Three microliters of the sample mix (0.5 mg/mL hOGA or $Ta$OGA) was added to freshly glow-discharged (45 s at 15 mA) grids, which were subsequently vitrified at 4 °C and 99% humidity for 3.5 s with blotting force 0. 0.6/1.0 μm 300 mesh AU grids (AuFlat, Protochips) were used for blotting and subsequent plunge freezing, which was carried out on a Vitrobot MarkIV plunge freezer (Thermo Fischer). Data were collected on a Titan Krios G3i microscope (EMBION Danish National Cryo-EM Facility – Aarhus node) operated at 300 kV equipped with a BioQuantum energy filter (energy slit width 20 eV) and K3 camera (Gatan). A nominal magnification of 130,000× was used, resulting in a pixel size of 0.647 Å$^2$/px with a total dose of 59.9 e$^-$/Å$^2$. The movies were fractionated into 52 frames (1.15 e$^-$/Å$^2$ per frame) at a dose rate of ~18 e$^-$/px/s and a 1.4 s exposure time per movie.

### Cryo-EM image processing

The images of the $Ta$OGA dataset were processed using the pipeline outlined in Supplementary Fig. 2. Micrographs were motion corrected[67] and CTF estimated[68] using CryoSPARC[42]. Particles were picked using a circular blob and aligned by 2D classification. Small subsets of the 2D classes were selected and used to generate ab initio volumes. Particles were primarily sorted alternating between heterogeneous refinement, with ab initio volumes created from selected 2D classes, and sorting by creating two ab initio volumes and carrying on

with the most well-defined particle stack. The particles were initially extracted in a bigger box to ensure that the pHAT domains were included. However, after realising the pHAT domains could not be resolved, the particles were re-extracted in a smaller box and finally refined in a non-uniform and local refinement[69].

The images of the hOGA dataset were processed using the pipeline outlined in Supplementary Fig. 5. The micrographs were motion corrected[67] and CTF estimated[68] using CryoSPARC[42]. Particles were picked using a circular blob and aligned by 2D classification. A small subset of the 2D classes was selected and used to reconstruct ab initio volumes. One of the ab initio volumes was used for template picking in all micrographs to extract particles. Particles from selected 2D classes were used as templates for re-extraction in a bigger box. Particles were sorted in 2D to remove junk, and reconstruction in several ab initio volumes that sorted dimeric from monomeric particles and junk. The particles were also sorted into four main conformations with these strategies. Finally, by using the "particle sets" tools, we separated particles overlapping in several conformations, to ensure particles in each conformation were unique. Each conformation was refined both in a non-uniform and a local refinement[69].

### Model building, refinement and validation

AlphaFold3 models were generated on February 3rd 2025 (human OGA; uniprot ID: O60502) and February 4th 2025 ($Ta$OGA; uniprot ID: A0A0D5X2Y8). The models were manually docked in the cryo-EM volumes and fitted to the map with geometry restraints using Namdinator[70]. Real-space refinement of the structures was done in Phenix[71], and model building and analysis were performed in Coot[66] (Supplementary Table 4). Model-to-map fit quality was assessed in part by evaluating helix placement within the density (Supplementary Fig. 7). Figures for the cryo-EM volumes and models were made in ChimeraX[46].

### SAXS data collection and analysis

SAXS was performed at the in-house facility at Aarhus University[72]. It is a modified Bruker AXS NanoSTAR that uses an Excillum Galium liquid metal jet X-ray source and a homebuilt scatterless pinhole (see, for example, European Patent EP13159569.6[73].), and is equipped with an automated sample handler, which injects the sample into a reusable quartz capillary, allowing samples and buffers for background subtraction to be measured in the same capillary. Water at 20 °C was used for absolute calibration. The resulting data are shown as the intensity $I(q)$ vs the modulus of the scattering vector $q$ in the plots. Resulting datasets were plotted in a log-log plot, and a Guinier plot of data and linear fit of $\ln(I(q))$ vs $q^2$ was performed to obtain values for $I(0)$ and the radius of gyration $R_g$. A dimensionless Kratky plot of $(R_g q)^2 I(q)/I(0)$ vs $(R_g q)$ was made to identify the influence of flexible parts. Indirect Fourier transformations were performed using the GNOM program of the ATSAS package[74] and BIFT[75]. Masses were estimated from the Porod volume $V_p$ and the correlation volume $V_c$ as determined by the ATSAS package. The agreement of the cryo-EM map and the SAXS data was checked using a recently published program, which scans a threshold value for the map, creates dummy atom models and determines the reduced chi-squared for each model, thus obtaining the model from the map that gives the best agreement with the SAXS data[49]. To obtain more information on the solution structure of hOGA, the SAXS data were further analysed using high-resolution models of the full-length hOGA obtained via Modeller software[50]. The scattering for this structure was calculated and compared to the SAXS data using a home-written program, which was described previously in refs. 76–79. In brief, the program uses an equivalent model approximation for the non-hydrogen atoms with a Gaussian form factor, adds a hydration layer around the protein, and calculates the SAXS intensity on an absolute scale using the Debye equation[80]. The program allows rigid-body optimisation of the structure employing soft connectivity

and excluded volume restraints. A structure factor for unspecific oligomers can also be fitted to account, for example, for the presence of a fraction of unspecific dimers[76]. We think that this is a good alternative to extrapolation to zero concentration, which introduces additional uncertainties, and to in-line measurements with size-exclusion chromatography (SEC-SAXS), where the technique leads to dilution and with laboratory-based SAXS to data of inferior quality[81].

The hOGA dimer was divided into eight bodies by identifying flexible and compact parts of the structure and considering that the structures should have some degree of freedom to relax during the optimisation. The agreement with the SAXS data was optimised by performing refinements of the starting structure using random translations and rotations. For each optimisation, ten runs were performed to investigate variations in the structure and to find the structure where the calculated scattering has the best agreement with the measured SAXS data. A satisfactory fit in terms of reduced chi-square, $\chi^2$, could not be obtained without including an oligomer structure factor.

### Determination of O-GlcNAc dyshomeostasis using a double fluorescence-labelled cell line

A previously engineered cell line incorporating endogenously labelled OGT-sfGFP/mScarlet3-OGA was used to measure effects on O-GlcNAc homoeostasis as a result of transfection of mTagBFP2-tagged exogenous hOGA and variants thereof[44,45]. The parental line was derived from E14-TG2a.IV (129/Ola) ES cells, a male cell line obtained from the MRC Centre for Regenerative Medicine, Institute for Stem Cell Research, University of Edinburgh[82]. No mycoplasma contamination was detected prior to the experiments. Briefly, $5.0 \times 10^5$ cells were transfected with 1.5 µg of DNA plasmid and 3 µL of Lipofectamine 2000 in a 12-well plate setting according to the manufacturer's instructions. 24 h post-transfection, the culture medium was replaced, and cells were harvested 48 h after transfection. Cells were fixed in warm 4% paraformaldehyde (PFA) with 4% sucrose for 10 min at room temperature, followed by washing with PBS. To block nonspecific binding and permeabilize the cells, samples were incubated in blocking buffer (5% bovine serum albumin (BSA) and 0.3% Triton X-100 in PBS) for 1 h at room temperature. Cells were incubated overnight at 4 °C in 100 µL of diluted blocking buffer (diluted to 1% BSA in PBS) containing 2 µL of RL2-Alexa Fluor 647-conjugated anti-O-GlcNAc antibody (Thermo Fisher Scientific; Cat. no. 51-9793-42; clone RL2; Lot 2786009). Following incubation, cells were washed with PBS and analysed by flow cytometry using a NovoCyte Quanteon 4025 flow cytometer. A detailed gating strategy is depicted in Supplementary Fig. 8.

### Reporting summary

Further information on research design is available in the Nature Portfolio Reporting Summary linked to this article.

## Data availability

The diffraction dataset together with the atomic coordinates for *Ta*pHAT has been deposited in the PDB under the code: 6R45. The atomic coordinates for the cryo-EM structures have been deposited under the PDB accession codes: 9QEP (*Ta*OGA) and 9QEN (hOGA). The electron microscopy density maps have been deposited in the Electron Microscopy Data Bank (EMBD) under the accession codes: EMD-53082 (*Ta*OGA) and EMD-53081 (hOGA). The SAXS data have been deposited at the Small Angle Scattering Biological Data Bank (SASBDB) under the accession code SASDXF3. Composite maps and models of conformation I–IV, as well as the two AlphaFold models used in this study, are found as supplementary files. In addition, the already published atomic coordinates PDB 7VVU, PDB 3ZJ0, PDB 4BMH, PDB 1WWZ, PDB 1Q2D and PDB 5VVO were used in this study. Size-exclusion chromatograms, DSF, FACS and SPR raw data files presented in this study can be downloaded as a source data file. Unless otherwise stated, all data

supporting the results of this study can be found in the article, supplementary, and source data files. Source data are provided with this paper.

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

## Acknowledgements

This work was funded by a Wellcome Trust Investigator Award (110061), a Novo Nordisk Fonden Laureate award (NNF21OC0065969) and a Villum Fonden Investigator (00054496) to D.M.F.v.A. Supported in part by the Danish Research Institute of Translational Neuroscience – DANDRITE of the Nordic-EMBL Partnership for Molecular Medicine and Lundbeckfonden. We would like to thank ESRF (beamline ID30A) for the synchrotron time. We are also grateful to Iva Hopkins-Navratilova and Tonia Aristotelous from the University of Dundee Drug Discovery Unit for their assistance in performing the SPR experiments. We also extend our gratitude to the FACS Core Facility at Aarhus University for their support. The authors would like to thank the staff of the EMBION cryo-EM facility and Jesper Karlsen for help using the EMCC facilities at Aarhus University for scientific computing.

## Author contributions

S.G.B. and D.M.F.v.A. conceived the study. S.B.H., S.G.B., T.D., T.B., and O.G.R. performed structural biology. S.G.B., O.G.R., J.S.P., K.L., and A.G. performed biochemical experiments. A.T.F. performed molecular biology. H.Y. performed flow cytometry and cellular experiments. S.G.B., S.B.H. and D.M.F.v.A. interpreted the data and wrote the manuscript with input from all authors.

## Competing interests

The authors declare no competing interests.
