## [Transparent Peer Review file · Nature Communications]

Multi-domain O-GlcNAcase structures reveal allosteric regulatory mechanisms

Corresponding Author: Professor Daan van Aalten

Version 0:

Reviewer comments:

Reviewer #1

(Remarks to the Author)

Page 7: Consistency of the homodimer claim

The conclusion that “eukaryotic pHAT domains may form homodimers” appears inconsistent with the preceding PISA analysis, which finds no conserved interface in other pHATs and mentions repeated crystallization failures. The text should acknowledge that the observed TapHAT homodimer may not be representative of metazoan pHATs without further supporting evidence.

Figure 3 and main text: Discrete cryo-EM subclasses versus underlying continuum

Only four conformations (I–IV) are presented, despite the raw 2D class averages suggesting a continuum of blurred pHAT density. Please clarify why these four states are interpreted as discrete rather than potential classification artefacts. Additionally, discuss how particle pruning (“particle-sets”) may introduce bias.

Page 14: Potential over-interpretation of W679 role

The discussion of Trp679 movement focuses on residue-level changes inferred from AlphaFold-guided models fitted into low-resolution, anisotropic density. Without mutational or biochemical validation, this mechanistic proposal may be overinterpreted. The argument would be stronger if supported by experimental data.

SAXS-specific comments:

Figure S6A: Instrumental error may be underestimated

Between $q \approx 0.20\text{--}0.30 \text{ \AA}^{-1}$, intensity fluctuations exceed the stated error bars, possibly due to detector noise or background subtraction issues.

Guinier fit: unreliability of initial data points

The first 4–6 points appear unreliable, potentially due to beam-stop shadowing, low-pixel statistics, or buffer mismatch. Excluding these may improve the fit and help explain the observed D_{max} differences between GNOM and BIFT.

Kratky plot interpretation

Aside from the final few data points—which are noisy and susceptible to experimental artefacts—the dimensionless Kratky plot is flat. Therefore, stating that it “does not reach a plateau” may be too strong. The data remain compatible with a multi-domain protein connected by flexible linkers.

$\rho(r)$ and D_{max} discrepancy

Did the authors try to manually adjust D_{max} in both GNOM and BIFT analyses? GNOM appears to truncate the distribution too early (intensity drops abruptly to zero), while BIFT underestimates the D_{max} (there seems to be no signal beyond 200 Å). On what basis is it claimed that GNOM is more sensitive to compactness and BIFT to oligomeric composition? (Does it just fit the story? Or is this interpretation based on a theoretical rationale?)

Model bias in rigid-body refinement

The SAXS model fitting uses rigid bodies and a structure factor to account for 60–70% “dimers of dimers.” Were methods that explicitly account for flexibility (e.g., EOM, SREFLEX), or polydispersity through fitting a mixture of models, considered?

Making the raw scattering data and final models publicly available (e.g., via SASBDB) would enable others to explore and test alternative interpretations.

Reviewer #2

(Remarks to the Author)

The authors showed that pseudo-histidine acetyl transferase (pHAT) domain associated with O-GlcNAc hydrolase (OGA) may play a role in allosteric regulation of O-GlcNAc homeostasis. The use of X-ray crystallography, Cryo-EM, and SAXS were employed to study the structural basis of said allostery. These complementary methods, along with previous studies, demonstrated that pHAT domains in conjunction with hOGA exhibits flexibility and multiple conformations.

Surface Plasmon Resonance (SPR) showed that acetyl-CoA was unable to bind metazoan proteins and thus ruled out any catalytic activity of pHAT, which further provided confirmation of pHAT's regulating O-GlcNAc hydrolysis/homeostasis. These structural mechanistic insights could provide alternative ways of treating/targeting diseases that result in intellectual disability caused by low O-GlcNAc levels.

The work provides an alternative method of therapeutic intervention based on allostery. Allostery has been known for sometime but in more recent times, interest has grown for targeting modulators. Understanding the structural architecture for such complexes/multidomain systems is crucial to treat diseases in a more sophisticated approach. Inability to control O-GlcNAc levels/homeostasis is the basis of many diseases, which are often neurodevelopmental in nature. The work will be of significance to the field.

The work supports the conclusions.

The authors stated that the *in crystallo* dimer is present in solution. They used size exclusion chromatography to verify that the dimer was in fact there. These statements should have been supported with more evidence. The authors, in Supplementary Fig. 1A, should state the molecular weight of the pHAT along with a higher resolution chromatogram, standards to definitely show the dimer exists in solution. If nothing else, an image of the SDS-PAGE should be provided. In the Materials & Methods crystallization portion, the authors should state how the suitable crystallization condition was found (previously known? Sparse matrix screen? Where there other conditions that also produced crystals?)

Table 1, The $CC_{1/2}$ (along with the I/σ) for the highest resolution shell isn't included in Table 1 nor in the deposition. The completeness is also just below 90%, the authors should provide an explanation as to why it is low, even if it's just because of it being in P1s. It is unknown whether the data were appropriately cut at an appropriate resolution.

If the authors include more details on crystallization, the detail provided in the methods for the work can be reproduced.

Reviewer #3

(Remarks to the Author)

The manuscript (Hansen et al.) investigates the structural and functional aspects of the O-GlcNAc hydrolase (OGA) enzymes from *Trichoplax adhaerens* and human, which regulates protein O-GlcNAcylation, a critical post-translational modification affecting various cellular processes. The study focuses on the poorly understood C-terminal pseudo-histone acetyltransferase (pHAT) domain of OGA. Using crystallography and cryo-electron microscopy, the manuscript determined the structures of the pHAT domain from *Trichoplax adhaerens* and multi-domain forms of both *T. adhaerens* and human OGA. They found that the pHAT domains contribute to O-GlcNAc homeostasis, forming catalytically inactive symmetric homodimers with a potential peptide-binding site. In human OGA (hOGA), the position of the pHAT domains influences the active site's environment through a conformational change involving a flexible arm region. These findings reveal allosteric regulatory mechanisms in OGA protein, suggesting that the pHAT domain mediates protein-protein interactions or allosteric regulation rather than catalyzing acetylation. Overall, this study highlights the importance of the pHAT domain in regulation of OGA's O-GlcNAc hydrolase function, providing some explanations for the allosteric regulation mechanisms of the conserved OGA from different species. Some concerns should be addressed.

Major points

1 The key point of the current study is the potential allosteric regulation of dimeric OGA by its multiple-conformation pHAT domain. The study showed the potential and transient interaction between the N-terminal O-GlcNAc hydrolase domain and the C-terminal pHAT domain by cryo-EM. However, the biochemical evidence is still lacking. Please try some biochemical assays (Western-blot based with protein substrate, or quantitative OGA activity assays using O-GlcNAcase assay kit with PNP-GlcNAc as substrate) to check the O-GlcNAc hydrolase activity of OGA with intact full-length protein and pHAT-less protein. The full-length WT protein and pHAT dimeric-deficient protein should be included for comparison.

2 The cooperativity between the pHAT and stalk domain of OGA may also extend into the active site. This phenomenon

should also happen when the substrates of OGA were bound. Is it possible to solve the cryo-EM structure of full-length OGA with certain well-studied protein substrate(s), this structure would enhance the conclusion reached by current study.

3 A recent complementary manuscript (<https://www.researchsquare.com/article/rs-6197257/v1>, Human O-GlcNAcase catalytic-stalk dimer anchors flexible histone binding domains) also did the cryo-EM study for human full-length OGA (OGA-L) but did not report the dimeric structure of pHAT domain and the potential allosteric regulation of human OGA, while they showed that the HAT domain of OGA-L specifically recognizes H3K36 and H4 acetylated nucleosome, the caveats of the two papers should be better clarified and discussed.

Minor points

1 One of the major conclusions of the current study is that the pHAT domains of eukaryotic OGAs form homodimers, please show the gel filtration profiles or the Analytical Ultracentrifugation (AUC) Assay for the full-length and the C-terminal pseudo-histone acetyltransferase (pHAT) domain of both *T. adhaerens* and human OGA, which will clearly demonstrate the dimeric arrangement of the conserved OGA protein.

2 Please biochemically validate the dimer interface of TapHAT, the mutations of which would disrupt dimer formation in solution. Can pHAT domain of human OGA form obligate dimer? Please show the predicted dimer (by AF3) of pHAT domain and full-length protein of human OGA.

3 in Supplementary Figure 3 and 4, please add a local mask to cover the monomer OGA only to improve the resolution for the full-length OGA, which may be better for modeling the HAT domain.

4 in Supplementary Figure 3 and 4, the raw micrographs looked in low-quality state and the particle looked ice-damaged, please replace them with better and representative ones. In addition, all the Guinier plots, Particle Viewing Distribution Plots and GSFSC Resolution Plots are in low-resolution mode and difficult to see. Please replace them using images with better-resolution.

Version 1:

Reviewer comments:

Reviewer #1

(Remarks to the Author)

We appreciate the revisions made to the manuscript and thank the authors for their clarifications. A few remarks remain:

1. Dimerization of pHAT domains from eukaryotic OGAs

I remain uncertain about the logical flow of the first section of the Results and Discussion ("The pHAT domains of eukaryotic OGAs form homodimers").

The authors present convincing evidence that the TapHAT domain forms a homodimer in solution, based on crystallographic data, PISA-calculated interface area, and chromatography and cross-linking experiments (Supplementary Fig. 1A). However, they also report that a PISA-based search for similar dimer interfaces across other eukaryotic OGA pHAT domains fails to identify conserved dimeric arrangements. This suggests that the TapHAT interface is not structurally conserved, and by itself does not support the conclusion that dimerization is a general feature of eukaryotic pHAT domains.

Nevertheless, the section is titled—and concludes with—the statement that “eukaryotic pHAT domains form homodimers.” If this conclusion is meant to derive from the PISA analysis, it appears inconsistent with the data presented. If, instead, it rests on independent experimental evidence, this is not clearly stated.

Notably, Supplementary Fig. 1A appears to show that the human pHAT domain elutes as a dimer under the same conditions. However, this is not mentioned in the main text at the point where the conclusion about eukaryotic dimerization is drawn. This omission is significant: if the human pHAT indeed forms dimers, that would provide critical support for the general conclusion—but it would also highlight a discrepancy between the structural conservation analysis (via PISA) and the experimental oligomeric state. Explicitly acknowledging this apparent contradiction would strengthen the clarity and rigor of the argument.

Finally, the idea that the pHAT domain forms a dimer because the catalytic core of hOGA does is not, in itself, a reliable inference. Domains within the same protein—especially when separated by flexible linkers or disordered regions—can have independent oligomerization behaviors. For instance, p53 forms a tetramer through its dedicated oligomerization domain, while its N-terminal transactivation and C-terminal regulatory domains remain monomeric. This illustrates that the oligomeric state of one domain does not necessarily dictate that of another within the same protein.

It is possible that I have misunderstood the intended basis for the conclusion. If so, a clearer distinction between structural conservation, experimental evidence, and functional inference would help resolve the ambiguity.

2. SAXS Analysis and Molecular Weight Interpretation

The SAXS analysis is generally well executed, and the recent clarifications are appreciated. One can, however, regret that SAXS measurements were not conducted across a concentration series, or ideally using SEC-SAXS. A concentration-dependent dataset could have allowed extrapolation to the pure dimer state, while SEC-SAXS would have directly isolated the dimeric species from higher-order assemblies. Either approach would have made the modeling of the dimer more robust, reducing the reliance on fitted components to account for dimer-of-dimer contamination. We acknowledge that such data are not always feasible to obtain, but this remains a limitation in interpreting the SAXS-derived structural model.

Additionally, a minor correction: in the SAXS section, the Porod and correlation volume molecular weights are still cited as 320 kDa. However, in the revised Table 2, these values have been updated and are now closer to 280 kDa, which more accurately reflects the presence of 60–70% dimer-of-dimer species, as discussed in the main text.

Reviewer #2

(Remarks to the Author)

The authors have addressed my original comments.

Reviewer #3

(Remarks to the Author)

The revised manuscript have been greatly improved and most of my comments have been convincingly addressed. I have no further comments.

Version 2:

Reviewer comments:

Reviewer #1

(Remarks to the Author)

The author has addressed the two points from the previous review. Thank you for the revisions.

REVIEWER COMMENTS

Reviewer #1 (Remarks to the Author):

Page 7: Consistency of the homodimer claim

The conclusion that “eukaryotic pHAT domains may form homodimers” appears inconsistent with the preceding PISA analysis, which finds no conserved interface in other pHATs and mentions repeated crystallization failures. The text should acknowledge that the observed TapHAT homodimer may not be representative of metazoan pHATs without further supporting evidence.

We thank the referee for their comment. To address the concern, we performed analytical size-exclusion chromatography followed by macromolecular crosslinking using glutaraldehyde. The new data show that the elution profile is consistent with a dimeric species. Furthermore, incubation with glutaraldehyde followed by denaturing SDS-PAGE reveals the presence of homodimers. These findings support our claim that both TapHAT and human pHAT form dimers in solution, suggesting that dimerization is a plausible oligomeric state for all isolated metazoan pHAT domains. This new data replaces panel (a) in Supplemental Figure 1.

Figure 3 and main text: Discrete cryo-EM subclasses versus underlying continuum

Only four conformations (I–IV) are presented, despite the raw 2D class averages suggesting a continuum of blurred pHAT density. Please clarify why these four states are interpreted as discrete rather than potential classification artefacts. Additionally, discuss how particle pruning (“particle-sets”) may introduce bias.

While 3D classifications revealed reconstructions for four distinct conformational states (I–IV) (Fig. 3c–f), we acknowledge that these likely represent only a subset of a broader conformational continuum. The blurred or elongated densities corresponding to the pHAT domains in the full particle set (Fig. 3a upper panels, 3b) — and the variability observed in lower-populated 2D classes (Fig. 3a lower panels) — indicate substantial inter-domain flexibility. The four resolved conformations were obtained through focused classification of the most populated particle subsets, allowing higher-resolution reconstructions. As such, they should be interpreted as higher populated states within a larger ensemble of dynamic conformations. The classification strategy relies on particle pruning and alignment decisions that can bias toward dominant states. We thus interpret conformations I–IV as representative poses along a flexible trajectory, rather than as rigid or fully discrete states. This view aligns with the continuous density seen when all particles are analyzed together, suggesting that hOGA samples a wide range of pHAT–catalytic core orientations in solution. We have updated the text on page 11 to emphasize this point.

Page 14: Potential over-interpretation of W679 role

The discussion of Trp679 movement focuses on residue-level changes inferred from AlphaFold-guided models fitted into low-resolution, anisotropic density. Without mutational or biochemical validation, this mechanistic proposal may be overinterpreted. The argument would be stronger if supported by experimental data.

We appreciate the reviewer’s insightful comment. To test this hypothesis, we substituted the tryptophan residue at position 679 with alanine (W679A) and transfected this OGA variant into our dual-fluorescent reporter cell line to assess its ability to disrupt O-GlcNAc homeostasis in comparison to wild-type OGA. The results demonstrate that the mutation does not affect the activity. We have rephrased the text on page 15–16 to focus on larger structural elements instead of movement of single amino acids.

SAXS-specific comments:

Figure S6A: Instrumental error may be underestimated

Between $q \approx 0.20\text{--}0.30 \text{ \AA}^{-1}$, intensity fluctuations exceed the stated error bars, possibly due to detector noise or background subtraction issues.

The programs for data processing use error propagation from the counting statistics, also taking into account normalization of the detector sensitivity and dark current, and there are no indications in

numerous applications that the equations have been implemented wrongly. The correct error bars are further supported by the fit quality of an Indirect Fourier Transformation, which gives a mean square residual (MRS) of 1.4, which is within the expected range for correctly estimated errors. We agree that there is an inaccuracy of the subtraction at the highest q reflected by the small increase in intensity in the last few points. However, the contribution of these points to the MSR or reduced χ^2 is very small due to the large error bars on these points. Note that the data has been re-binned to improve the statistics at high- q and thus provide a greater challenge to the models. Despite this, the rigid-body optimized model have χ^2 value in the range 2.9-6.5 as already mentioned in the manuscript on page 14.

Guinier fit: unreliability of initial data points

The first 4–6 points appear unreliable, potentially due to beam-stop shadowing, low-pixel statistics, or buffer mismatch. Excluding these may improve the fit and help explain the observed D_{\max} differences between GNOM and BIFT.

We thank the reviewer for noticing these data points. The data were already corrected for beamstop shadowing effects as described in (Pedersen, J. S. (2004) A flux-and background-optimized version of the NanoSTAR small-angle X-ray scattering camera for solution scattering. *Applied Crystallography*, 37(3), 369-380.). But we agree that there is a problem with these data points, however, it is most likely due to poor background subtraction, i.e. that the capillary was cleaner for the sample than for the buffer. These points were not included in the Guinier fit as revealed by the residual plot. We have now removed these data points completely from the plot and have updated Supplementary Fig. 6.

Kratky plot interpretation

Aside from the final few data points—which are noisy and susceptible to experimental artefacts—the dimensionless Kratky plot is flat. Therefore, stating that it “does not reach a plateau” may be too strong. The data remain compatible with a multi-domain protein connected by flexible linkers.

Unfortunately, there was an error in the sentence rendering it contradictory. We agree with the reviewer that the data do reach a plateau at high q , compatible with the presence of loops. We have changed the text on page 13 accordingly.

$p(r)$ and D_{\max} discrepancy

Did the authors try to manually adjust D_{\max} in both GNOM and BIFT analyses? GNOM appears to truncate the distribution too early (intensity drops abruptly to zero), while BIFT underestimates the D_{\max} (there seems to be no signal beyond 200 Å). On what basis is it claimed that GNOM is more sensitive to compactness and BIFT to oligomeric composition? (Does it just fit the story? Or is this interpretation based on a theoretical rationale?)

We appreciate the feedback on this analysis. We have checked the GNOM inversion with the ATSAS version 4 and default values gives a D_{\max} of around 190 Å. We have also checked the BIFT using the server at <https://somo.chem.utk.edu/bayesapp/> and it also gives D_{\max} of 190. So apparently there are no discrepancies with the newest version of the programs. We have changed Supplementary Fig. 6 (shown above) and the text on page 13 accordingly and no longer make suggestions about specific applicability of the two approaches.

Model bias in rigid-body refinement

The SAXS model fitting uses rigid bodies and a structure factor to account for 60–70% “dimers of dimers.” Were methods that explicitly account for flexibility (e.g., EOM, SREFLEX), or polydispersity through fitting a mixture of models, considered? Making the raw scattering data and final models publicly available (e.g., via SASBDB) would enable others to explore and test alternative interpretations.

The models did account for flexibility through the rigid-body refinement, however, ensemble methods, like those mentioned, were not applied since the models gave quite low χ^2 values (2.9-6.5) without this. Including further degrees of freedom in the optimization would lead to the risk of overfitting the

data. The modelling approach does allow some ensemble averaging of the 'dimer-of-dimer' structure, as the oligomerization factor describes unspecific aggregation (see Bærentsen, R. L., Nielsen, S. V., Skjærning, R. B., Lyngsø, J., Bisiak, F., Pedersen, J. S., Gerdes, K., Sørensen, M. A. & Brodersen, D. E. (2023). *eLife* 12, <https://doi.org/10.7554/eLife.90400.2>for details.). So effectively both the dimer and the dimer-of-dimer are optimized simultaneously during the rigid-body refinement, which also includes the ratio of between the two species as well as the average separation of the dimers within a dimer-of-dimers. We will make the data and model available on SASBDB.

Reviewer #2 (Remarks to the Author):

The authors showed that pseudo-histidine acetyl transferase (pHAT) domain associated with O-GlcNAc hydrolase (OGA) may play a role in allosteric regulation of O-GlcNAc homeostasis. The use of X-ray crystallography, Cryo-EM, and SAXS were employed to study the structural basis of said allostery. These complementary methods, along with previous studies, demonstrated that pHAT domains in conjunction with hOGA exhibits flexibility and multiple conformations.

Surface Plasmon Resonance (SPR) showed that acetyl-CoA was unable to bind metazoan proteins and thus ruled out any catalytic activity of pHAT, which further provided confirmation of pHAT's regulating O-GlcNAc hydrolysis/homeostasis. These structural mechanistic insights could provide alternative ways of treating/targeting diseases that result in intellectual disability caused by low O-GlcNAc levels.

The work provides an alternative method of therapeutic intervention based on allostery. Allostery has been known for sometime but in more recent times, interest has grown for targeting modulators. Understanding the structural architecture for such complexes/multidomain systems is crucial to treat diseases in a more sophisticated approach. Inability to control O-GlcNAc levels/homeostasis is the basis of many diseases, which are often neurodevelopmental in nature.

The work will be of significance to the field.

The work supports the conclusions.

The authors stated that the *in crystallo* dimer is present in solution. They used size exclusion chromatography to verify that the dimer was in fact there. These statements should have been supported with more evidence. The authors, in Supplementary Fig. 1A, should state the molecular weight of the pHAT along with a higher resolution chromatogram, standards to definitely show the dimer exists in solution. If nothing else, an image of the SDS-PAGE should be provided. In the Materials & Methods crystallization portion, the authors should state how the suitable crystallization condition was found (previously known? Sparse matrix screen? Where there other conditions that also produced crystals?)

Table 1, The $CC_{1/2}$ (along with the $I/\sigma I$) for the highest resolution shell isn't included in Table 1 nor in the deposition. The completeness is also just below 90%, the authors should be provide an explanation as to why it is low, even if it's just because of it being in P1s. It is unknown whether the data were appropriately cut at an appropriate resolution.

If the authors include more details on crystallization, the detail provided in the methods for the work can be reproduced.

We thank the referee for the constructive comments. As requested, we have now included an improved version of the size-exclusion chromatography profile, incorporating molecular weight markers for reference. Additionally, we performed a crosslinking experiment using glutaraldehyde, which confirms the presence of pHAT homodimers in solution. The new data replaces Panel (a) in supplemental figure 1.

Regarding the crystallization conditions, they were initially identified using the INDEX 1 and 2 screens (Hampton Research). Although several promising hits were observed, only one condition yielded crystals of sufficient size and quality. Further optimisation of this condition was necessary to obtain crystals suitable for diffraction. Material and methods have been updated accordingly.

As requested, the missing CC1/2 (0.930) and I/ σ I (3.0) values have now been included in Table 1.

We acknowledge the reviewer's observation that the completeness of the dataset (89%) falls slightly below 90%. This is attributable to the use of a non-symmetric space group (P1). Initial indexing at the beamline suggested a monoclinic C2/m (#12) space group, and the data collection strategy was based on this assignment, resulting in the acquisition of 1114 images with an oscillation range of 0.15° per image (totaling 167°). Upon reprocessing the dataset in the lab, the correct space group was identified as triclinic P1 (#1), which ideally requires 180° of data collection. Consequently, we were 13° short. Nevertheless, the resulting completeness of 89% in P1 is acceptable, as values above 85% are generally sufficient for successful structure determination in such low-symmetry space groups.

As expected, the overall multiplicity of the dataset was modest (1.6), but this remains within the acceptable range for molecular replacement. Importantly, both the I/ σ I (3.0) and CC1/2 (0.930) values indicate strong signal-to-noise and excellent internal consistency, supporting the quality of the data and justifying our decision to proceed with structure determination.

Regarding the resolution cut-off, this was initially determined through manual inspection of the first diffraction images during beamline data collection. This approach may have slightly underestimated the usable resolution, especially given the high sensitivity and accuracy of detectors such as the PILATUS 6M. We agree with the reviewer that the I/ σ I value of 3.0 suggests the potential to extend the resolution cut-off further. However, the dataset processed to 1.78 Å resolution was sufficient to solve the TaPHAT 3D structure with high accuracy.

Reviewer #3 (Remarks to the Author):

The manuscript (Hansen et al.) investigates the structural and functional aspects of the O-GlcNAc hydrolase (OGA) enzymes from *Trichoplax adhaerens* and human, which regulates protein O-GlcNAcylation, a critical post-translational modification affecting various cellular processes. The study focuses on the poorly understood C-terminal pseudo-histone acetyltransferase (pHAT) domain of OGA. Using crystallography and cryo-electron microscopy, the manuscript determined the structures of the pHAT domain from *Trichoplax adhaerens* and multi-domain forms of both *T. adhaerens* and human OGA. They found that the pHAT domains contribute to O-GlcNAc homeostasis, forming catalytically inactive symmetric homodimers with a potential peptide-binding site. In human OGA (hOGA), the position of the pHAT domains influences the active site's environment through a conformational change involving a flexible arm region. These findings reveal allosteric regulatory mechanisms in OGA protein, suggesting that the pHAT domain mediates protein-protein interactions or allosteric regulation rather than catalyzing acetylation. Overall, this study highlights the importance of the pHAT domain in regulation of OGA's O-GlcNAc hydrolase function, providing some explanations for the allosteric regulation mechanisms of the conserved OGA from different species. Some concerns should be addressed.

Major points

1 The key point of the current study is the potential allosteric regulation of dimeric OGA by its multiple-conformation pHAT domain. The study showed the potential and transient interaction between the N-terminal O-GlcNAc hydrolase domain and the C-terminal pHAT domain by cryo-EM. However, the biochemical evidence is still lacking. Please try some biochemical assays (Western-blot based with protein substrate, or quantitative OGA activity assays using O-GlcNAcase assay kit with PNP-GlcNAc as substrate) to check the O-GlcNAc hydrolase activity of OGA with intact full-length protein and pHAT-less protein. The full-length WT protein and pHAT dimeric-deficient protein should be included for comparison.

We appreciate these suggestions and agree that additional experiments are needed. Molecules such as PNP-GlcNAc and 4MU-GlcNAc have been instrumental in assessing kinetic differences among GH84

enzymes and in evaluating the effects of active site mutations. However, due to their small size, these substrates are not well suited to probe structural changes or mutations outside the active site. Despite this limitation, the catalytic consequences of removing the pHAT domain from OGA have already been examined using 4MU-GlcNAc and PNP-GlcNAc. For instance, a 2017 study accompanying the crystal structure of the human OGA catalytic core plus stalk domains reported that a pHAT-truncated version of OGA exhibited slightly enhanced activity against 4MU-GlcNAc compared to the wild-type enzyme, though the authors concluded that the overall difference was not significant (Nature Chemical Biology, 2017, 13). These findings were later corroborated by an independent group, which reached similar conclusions by comparing another pHAT-less truncation with wild-type OGA (Cell Death and Disease, 2021, 12:622).

To overcome the limitations of small-molecule probes in detecting the effects of mutations outside the active site, we measured global O-GlcNAcylation levels in cells using the RL2 anti-O-GlcNAc antibody in a flow cytometry assay. Our results show that overexpression of a pHAT-less OGA construct significantly decreases the RL2 signal compared to cells expressing wild-type OGA. This reduction in global O-GlcNAc levels correlates with the increased catalytic activity of the truncated OGA reported in the 2017 study, strongly suggesting that removal of the pHAT domain enhances OGA's substrate promiscuity. We have updated the text and present the new data on the Supplementary Fig. 5.

2 The cooperativity between the pHAT and stalk domain of OGA may also extend into the active site. This phenomenon should also happen when the substrates of OGA were bound. Is it possible to solve the cryo-EM structure of full-length OGA with certain well-studied protein substrate(s), this structure would enhance the conclusion reached by current study.

We agree with the reviewer that a crosstalk between the catalytic core and the pHAT domain likely plays a role in regulating OGA's interaction with its diverse protein substrates. However, substrates capable of engaging both the catalytic core and the pHAT domain simultaneously have yet to be identified. The discovery of such interactors, as well as the structural trapping of OGA-substrate macromolecular complexes and their functional characterisation and validation, represents an exciting but distinct line of investigation that falls beyond the scope of the current study.

3 A recent complementary manuscript (<https://www.researchsquare.com/article/rs-6197257/v1>, Human O-GlcNAcase catalytic-stalk dimer anchors flexible histone binding domains) also did the cryo-EM study for human full-length OGA (OGA-L) but did not report the dimeric structure of pHAT domain and the potential allosteric regulation of human OGA, while they showed that the HAT domain of OGA-L specifically recognizes H3K36 and H4 acetylated nucleosome, the caveats of the two papers should be better clarified and discussed.

As requested, we have referred to the preprint version of the mentioned work on page 18.

Minor points

1 One of the major conclusions of the current study is that the pHAT domains of eukaryotic OGAs form homodimers, please show the gel filtration profiles or the Analytical Ultracentrifugation (AUC) Assay for the full-length and the C-terminal pseudo-histone acetyltransferase (pHAT) domain of both *T. adhaerens* and human OGA, which will clearly demonstrate the dimeric arrangement of the conserved OGA protein.

We thank the referee for the helpful comments. As requested, we have replaced the original panel (a) in supplemental figure 1 with a revised version that includes molecular weight markers alongside *TapHAT* and human pHAT. In addition, we have validated these results by performing a crosslinking experiment using glutaraldehyde, confirming the presence of homodimers for both *TapHAT* and human pHAT in solution.

2 Please biochemically validate the dimer interface of *TapHAT*, the mutations of which would disrupt

dimer formation in solution. Can pHAT domain of human OGA form obligate dimer? Please show the predicted dimer (by AF3) of pHAT domain and full-length protein of human OGA.

As requested, we mutate all the identified residues to alanine and measure their elution profiles and their thermal stability by DSF. Despite observing a decrease in the thermal stability in some of the mutants the dimer formation was not affected, therefore we concluded that the TapHAT dimer might be stabilized only by the intermolecular interaction between each monomer b1 strands. Text has been modified to include those finding on page 7 and data showing the DSF profiles is now included in Supplementary figure 1, panel (c). As requested, the AlphaFold3 model of the full OGA protein is now included in Supplemental Fig. 2.

3 in Supplementary Figure 3 and 4, please add a local mask to cover the monomer OGA only to improve the resolution for the full-length OGA, which may be better for modeling the HAT domain.

Local masks for the OGA monomer have been generated. The global resolution decreased slightly and both local resolution and local map quality did not improve significantly.

4 in Supplementary Figure 3 and 4, the raw micrographs looked in low-quality state and the particle looked ice-damaged, please replace them with better and representative ones. In addition, all the Guinier plots, Particle Viewing Distribution Plots and GSFSC Resolution Plots are in low-resolution mode and difficult to see. Please replace them using images with better-resolution.

The micrographs are replaced in Supplementary Figure 3 and 4, and all SAXS related plots are replaced with better resolved images.

Response to the referees:

REVIEWER COMMENTS

Reviewer #1 (Remarks to the Author):

We appreciate the revisions made to the manuscript and thank the authors for their clarifications. A few remarks remain:

1. Dimerization of pHAT domains from eukaryotic OGAs

I remain uncertain about the logical flow of the first section of the Results and Discussion ("The pHAT domains of eukaryotic OGAs form homodimers").

The authors present convincing evidence that the TapHAT domain forms a homodimer in solution, based on crystallographic data, PISA-calculated interface area, and chromatography and cross-linking experiments (Supplementary Fig. 1A). However, they also report that a PISA-based search for similar dimer interfaces across other eukaryotic OGA pHAT domains fails to identify conserved dimeric arrangements. This suggests that the TapHAT interface is not structurally conserved, and by itself does not support the conclusion that dimerization is a general feature of eukaryotic pHAT domains.

Nevertheless, the section is titled—and concludes with—the statement that “eukaryotic pHAT domains form homodimers.” If this conclusion is meant to derive from the PISA analysis, it appears inconsistent with the data presented. If, instead, it rests on independent experimental evidence, this is not clearly stated.

We agree with the referee that, despite the presence of in-silico data, our experimental results (analytical size exclusion chromatography and crosslinking experiments) support the oligomerisation patterns only of TapHAT and human pHAT in isolation. Accordingly, we have revised the text and updated the section title to reflect this, removing any suggestion that dimerisation is a general “eukaryotic feature.”

Notably, Supplementary Fig. 1A appears to show that the human pHAT domain elutes as a dimer under the same conditions. However, this is not mentioned in the main text at the point where the conclusion about eukaryotic dimerization is drawn. This omission is significant: if the human pHAT indeed forms dimers, that would provide critical support for the general conclusion—but it would also highlight a discrepancy between the structural conservation analysis (via PISA) and the experimental oligomeric state. Explicitly acknowledging this apparent contradiction would strengthen the clarity and rigor of the argument.

We have now added a sentence to the main text stating that both TapHAT and human pHAT domains in isolation elute as dimers in our analytical size-exclusion chromatography experiments, and this observation was confirmed in both cases by crosslinking. In addition, we have clarified that the lack of structural homologs identified by PISA suggests that this dimeric arrangement has not been previously characterised.

To further explore this discrepancy, we have expanded the comparative analysis to include the monomeric OgHAT and SspHAT domains, as well as the dimeric PhHAT and TapHAT. This extended comparison has been incorporated into the text, and Supplementary Fig. 1b has been updated to include the sequence and structural alignment of the β 1 strands of OgHAT, SspHAT, PhHAT, and TapHAT.

Finally, the idea that the pHAT domain forms a dimer because the catalytic core of hOGA does is not,

in itself, a reliable inference. Domains within the same protein—especially when separated by flexible linkers or disordered regions—can have independent oligomerization behaviors. For instance, p53 forms a tetramer through its dedicated oligomerization domain, while its N-terminal transactivation and C-terminal regulatory domains remain monomeric. This illustrates that the oligomeric state of one domain does not necessarily dictate that of another within the same protein.

It is possible that I have misunderstood the intended basis for the conclusion. If so, a clearer distinction between structural conservation, experimental evidence, and functional inference would help resolve the ambiguity.

We agree with the referee's statement that "the oligomeric state of one domain does not necessarily dictate that of another within the same protein." In response, we have revised the relevant section of the manuscript to clarify that the dimerisation observed for the pHAT when the domain is expressed in isolation is not related to the oligomeric state of the catalytic core.

This conclusion is further supported by our cryoEM data, which show that within the full-length human OGA protein, the pHAT domain adopts several conformational states, including both monomeric and dimeric arrangements, while the catalytic core consistently remains an obligate dimer.

2. SAXS Analysis and Molecular Weight Interpretation

The SAXS analysis is generally well executed, and the recent clarifications are appreciated. One can, however, regret that SAXS measurements were not conducted across a concentration series, or ideally using SEC-SAXS. A concentration-dependent dataset could have allowed extrapolation to the pure dimer state, while SEC-SAXS would have directly isolated the dimeric species from higher-order assemblies. Either approach would have made the modeling of the dimer more robust, reducing the reliance on fitted components to account for dimer-of-dimer contamination. We acknowledge that such data are not always feasible to obtain, but this remains a limitation in interpreting the SAXS-derived structural model.

We agree that SEC-SAXS at synchrotrons-bases instruments is an ideal technique for separating monomer scattering from that of the oligomers. However, the technique is not so easily used for laboratory-based SAXS as the flux is significantly lower and this in combination with the dilution that occurs in the separation leads to data of inferior quality (Bucciarelli S, Midtgaard SR, Nors Pedersen M, Skou S, Arleth L, Vestergaard B. Size-exclusion chromatography small-angle X-ray scattering of water soluble proteins on a laboratory instrument. *J Appl Crystallogr.* 2018 Nov 9;51(Pt 6):1623-1632. doi: 10.1107/S1600576718014462. PMID: 30546289; PMCID: PMC6276278.). In order to obtain data of sufficient quality that challenges our models, we used a higher concentration data set and instead used the modelling of the oligomerization effects as we have introduced it (Bærentsen, R. L., Nielsen, S. V., Skjerning, R. B., Lyngsø, J., Bisiak, F., Pedersen, J. S., Gerdes, K., Sørensen, M. A. & Brodersen, D. E., Structural basis for kinase inhibition in the tripartite *E. coli* HipBST toxin-antitoxin system. 6 nov. 2023, *l: eLife.* 12, 20 s., RP90400.) and used it in the present work. As oligomerisation only influences the data at low q , we believe that this is a good alternative approach for laboratory-based SAXS. We note that although widely used in the past in particular in combination with light scattering, extrapolation to zero concentration gives additional uncertainties on the resulting data due to the inherent approximations involved in the procedure.

We have in the method section of the manuscript included the sentence:

"We think that this is a good alternative to extrapolation to zero concentration, which introduces additional uncertainties, and to in-line measurements with size-exclusion chromatography (SEC-SAXS), where the technique leads to dilution and with laboratory-based SAXS to data of inferior quality (reference: Bucciarelli S, Midtgaard SR, Nors Pedersen M, Skou S, Arleth L, Vestergaard B. Size-exclusion chromatography small-angle X-ray scattering of water soluble proteins on a laboratory

instrument. J Appl Crystallogr. 2018 Nov 9;51(Pt 6):1623-1632. doi: 10.1107/S1600576718014462. PMID: 30546289; PMCID: PMC6276278.)”

Additionally, a minor correction: in the SAXS section, the Porod and correlation volume molecular weights are still cited as 320 kDa. However, in the revised Table 2, these values have been updated and are now closer to 280 kDa, which more accurately reflects the presence of 60–70% dimer-of-dimer species, as discussed in the main text.

We apologize for this and have now corrected it.

Reviewer #2 (Remarks to the Author):

The authors have addressed my original comments.

Reviewer #3 (Remarks to the Author):

The revised manuscript have been greatly improved and most of my comments have been convincingly addressed. I have no further comments.